# Unravelling rechargeable zinc-copper batteries by a chloride shuttle in a biphasic electrolyte

Chen Xu[1], Chengjun Lei[1], Jinye Li[1], Xin He[1], Pengjie Jiang[1], Huijian Wang[1], Tingting Liu[1] & Xiao Liang [1]✉

The zinc-copper redox couple exhibits several merits, which motivated us to reconstruct the rechargeable Daniell cell by combining chloride shuttle chemistry in a zinc chloride-based aqueous/organic biphasic electrolyte. An ion-selective interface was established to restrict the copper ions in the aqueous phase while ensuring chloride transfer. We demonstrated that the copper-water-chloro solvation complexes are the descriptors, which are predominant in aqueous solutions with optimized concentrations of zinc chloride; thus, copper crossover is prevented. Without this prevention, the copper ions are mostly in the hydration state and exhibit high spontaneity to be solvated in the organic phase. The zinc-copper cell delivers a highly reversible capacity of 395 mAh g$^{-1}$ with nearly 100% coulombic efficiency, affording a high energy density of 380 Wh kg$^{-1}$ based on the copper chloride mass. The proposed battery chemistry is expandable to other metal chlorides, which widens the cathode materials available for aqueous chloride ion batteries.

The worldwide exploitation of alternative renewable energies (wind and solar) has led to an increase in demand for the storage of electrical energy, particularly for advanced batteries that show practical potential for grid-scale applications[1,2]. Although rechargeable lithium-ion batteries are the focus of the current energy storage market/industry, ubiquitous issues, including safety, toxicity, and resource limitation, have impeded their large-scale and/or security-critical applications[3,4]. Alternatively, aqueous zinc batteries combine several advantages, including intrinsic safety, high energy density, and abundant resources, and have been intensively studied recently as the most compelling substitutes for large-scale energy storage[5,6]. Although encouraging, aqueous zinc batteries are challenged by the sluggish intercalation/deintercalation kinetics of metal oxide cathodes due to the large electron density of Zn$^{2+}$ ions; therefore, only a limited rate capability and low capacity can be delivered[7]. The development of novel battery systems that incorporate aqueous zinc chemistry and new operating mechanisms would be a solution for the application of aqueous zinc batteries[8].

The historic Daniell cell, invented by the British chemist John Frederic Daniell in 1836, is popularly known as the zinc-copper battery,

which integrates the merits of high theoretical capacity and abundant resources for both Zn and Cu. Several modern battery designs have been developed to satisfy different application scenarios[9,10]; however, the Daniell cell is regarded as a primary cell due to the irreversibility (in neutral solution) or passivation of Cu$_2$O formed during discharge (in alkaline solution)[11]. It is currently used in chemistry curricula to demonstrate the battery working principle by using a salt bridge to facilitate SO$_4^{2-}$ anion shuttling between the cathode and anode. The total electrochemical reaction on the cathode and anode could be interpreted as [Zn(s) + Cu$^{2+}$(aq) ⇌ Zn$^{2+}$(aq) + Cu(s)], for which the crossover of the copper ion leads to a direct chemical reaction in the absence of a salt bridge. Efforts have been dedicated to making the Zn−Cu Daniell battery reversible, in which ion-exchange membrane/ ceramics are used to prevent Cu crossover in the neutral electrolyte[12–14], or transfer the redox electrochemistry to hydroxyl (OH$^-$) involved precipitation process to minimize the copper ion dissolution in alkaline electrolyte[15]. The incorporation of CuO and Bi$_2$O$_3$ could further mitigate Cu$_2$O passivation for the alkaline Zn−CuO battery, which affords a reversible Zn−CuO alkaline battery[11].

[1]State Key Laboratory of Chem/Biosensing and Chemometrics, Advanced Catalytic Engineering Research Center of the Ministry of Education, College of Chemistry and Chemical Engineering, Hunan University, Changsha, PR China. ✉e-mail: xliang@hnu.edu.cn

Nevertheless, the ion-exchange membrane approach significantly increases the cost of the technology, while the hydroxyl strategy results in a large polarization during charge and discharge due to the slow kinetics of the solid–solid conversion. These approaches provide valuable inspiration for the promising design of rechargeable Zn–Cu batteries, namely, the dissolution and crossover of copper species should be dialectically considered.

The electrochemistry of the Daniell cell (in neutral solution) is analogous to that of the typical chloride ion battery (CIB), which alternatively uses a chloride ion shuttle between the metal chloride/metal electrochemical couple[16,17]. In addition to the large variety of abundant chloride-containing electrode materials, CIB technology is attractive due to the wide array of possible electrochemical couples with high theoretical energy densities[18,19]. Moreover, the fast chloride anion transfer in the electrolyte solution provides relatively fast reaction kinetics for the electrochemical reaction in theory; the chloride ion radius in solvated cations is significantly smaller than that of the solvated cations. In principle, CIBs are also challenged by poor cycling stability due to the dissolution of metal chlorides[16,20]. Therefore, anion hosts such as metal oxychlorides[21,22], layered double hydroxides[23,24], and chloride ion-doped polymers[25,26] have been developed as cathode materials, but the poor structural stability of these frameworks has yet to be resolved[19].

In this report, we combine the advantages of the Zn–Cu redox and the glamorous chloride anion shuttle to develop a rechargeable historic Daniell cell. It was achieved by exploiting an aqueous/organic biphasic electrolyte, which established an ion selective interface that allows the restriction of the copper ions in the aqueous phase, along with chloride ions, which served as the charge carrier between the two phases to keep electrical neutrality. This biphasic electrolyte is composed of $ZnCl_2$ aqueous solution and a $Tf_2N$-based ionic liquid, in which the cathode is located in the aqueous phase while the zinc anode is immersed in the organic phase. The crossover of Cu ions across the interface of the biphasic electrolyte is strongly correlated to the coordination structure of Cu ions, which is rationally tunable according to the $ZnCl_2$ concentration in the aqueous phase. We demonstrated that the copper-water-chloro complexes are the descriptors that inhibits the occurrence of Cu in the organic phase, which is dominant in the aqueous solution with > 15 m $ZnCl_2$; otherwise, the copper ions are mostly in their hydration states with spontaneity to be solvated in the organic phase. This enables a large $Cu^{2+}$ distribution ratio of 2700 between the 15 m $ZnCl_2$ aqueous phase and organic phase, which affords highly reversible copper electrochemistry with stepwise redox reactions between $Cu^{II} – Cu^{I} – Cu^{0}$. The solution solvation structure-related ion selective interface of the biphasic electrolyte in this work could greatly alleviate the active material crossover and provide fast conversion kinetics. In addition, utilization of the active material was improved because the active material species were partially solubilized in the aqueous phase. We further demonstrated that this strategy of eliminating metal ions in the organic phase is expandable to iron chloride, nickel chloride, and vanadium oxides, representing a robust approach to improve the performance of chloride shuttle batteries.

## Results

### The biphasic electrolytes for the rechargeable Zn–Cu battery

Simply combining the $ZnSO_4$ aqueous electrolyte and the super P-$CuCl_2$ cathode renders fast capacity decay, which is attributed to the serious copper ion dissolution and its rapid chemical reactions with the zinc anode (Fig. 1a). An "ion-selective" interface is essential for preventing the crossover of electroactive metal cations between the cathode and anode. The aqueous-organic biphasic system, in which two phases with different compositions coexist as separate liquids, is designed to build such an ion-selective interface. The $Tf_2N$-based ionic liquids were selected as the organic phase due to their high

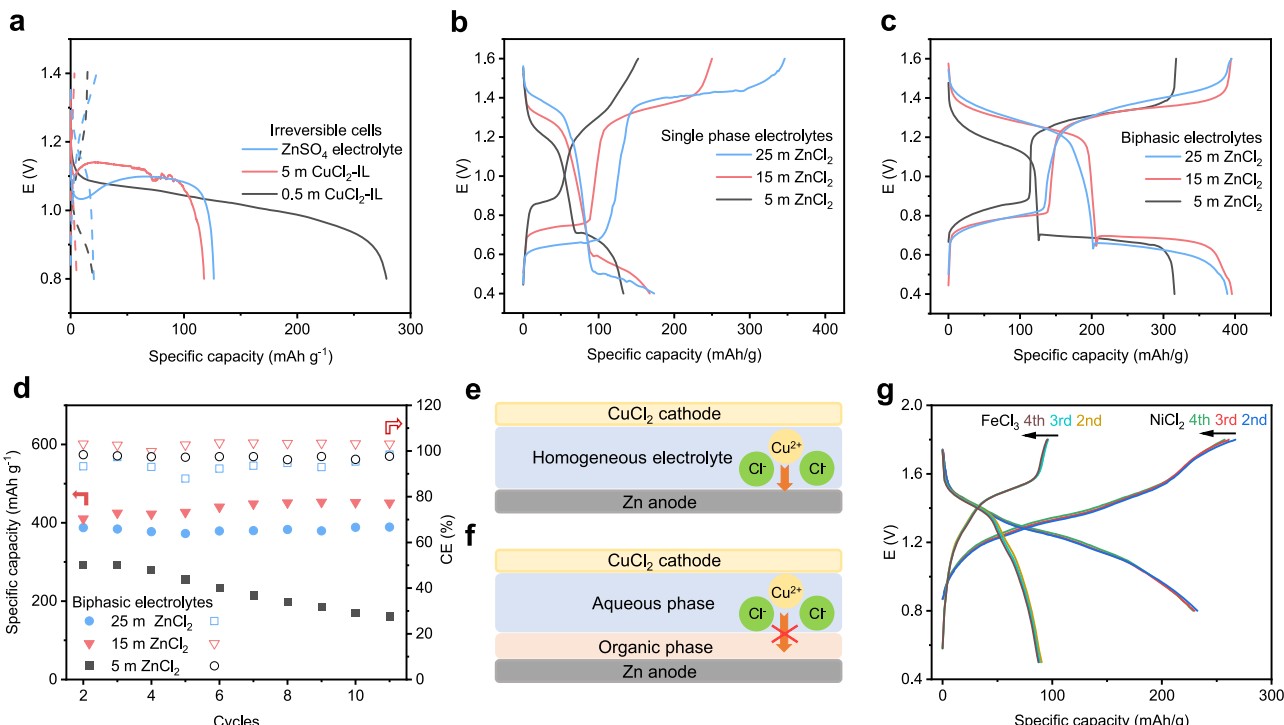

**Fig. 1 | The performance of Zn–Cu batteries based on various electrolytes.** **a** Irreversible cells using $ZnSO_4$ aqueous electrolyte or $CuCl_2$ aqueous solution/[bmmim][$Tf_2N$] biphasic electrolyte; solid lines are the first cycle, and dashed lines are the second cycle. **b** The fast degradation cells with the single phase aqueous electrolytes with different $ZnCl_2$ concentrations, **c** and the reversible cells with $ZnCl_2$-based biphasic electrolyte. **d** Capacity retention and coulombic efficiencies of Zn–Cu batteries based on $ZnCl_2$-based biphasic electrolytes. Schematic illustrations of Zn–Cu cells based on **e** a single-phase electrolyte and **f** a biphasic electrolyte. **g** Voltage profiles of the $FeCl_3$ and $NiCl_2$ cathodes in the $ZnCl_2$-based biphasic electrolyte. The current density in these studies was 200 mA g$^{-1}$ (0.5 C).

hydrophobicity and rapid ionic conductivity but poor dissolving ability for hydrated metal chloride. Although the neat $CuCl_2$ aqueous solution exhibited ready phase separation with the ionic liquid, the Zn–Cu cell based on this biphasic electrolyte again showed fast capacity degradation (carbon cloth was used as the current collector) (Fig. 1a). EDS results showed that a significant amount of Cu was deposited on the zinc anode after cycling, indicating the crossover of the Cu cations in such biphasic electrolytes (Supplementary Fig. 1).

The proposed immiscible biphasic electrolyte formula is composed of $ZnCl_2$ aqueous solution (>15 m, molality) and a $Tf_2N$-based ionic liquid phase with large hydrophobic cations (named $ZnCl_2$-based biphasic electrolyte), in which the dissolved electroactive $CuCl_2$ is automatically restricted in the aqueous phase. Due to the strong water repulsion of the hydrophobic anion and cation of the organic phase, the water-favorable metal ions show a distinct distribution in the biphasic electrolyte, most likely in the aqueous phase, which will be discussed in the next section. Note that low CE and rapid capacity fading are observed in the Zn–Cu batteries based on the single phase aqueous electrolyte with 5 m $ZnCl_2$, 15 m $ZnCl_2$, and 25 m $ZnCl_2$ (Fig. 1b). Figure 1c depicts the typical voltage profile of the Zn–Cu cell based on the $ZnCl_2$-based biphasic electrolyte, showing the continuous reduction of the copper ions with improved reversibility.

There are two distinct voltage plateaus for the copper redox reaction, in which the higher one corresponds to $Cu^{II}$ – $Cu^{I}$ conversion and the lower one is assigned to $Cu^{I}$ – $Cu^{0}$[27]. This stepwise reduction of copper ions is also observed in the three-electrode CV curves (Supplementary Fig. 2), showing two pairs of redox waves that are consistent with the voltage profiles. Moreover, the potential of the redox peak is strongly correlated to the $ZnCl_2$ concentration, among which a high $ZnCl_2$ concentration of the biphasic electrolyte would increase the $Cu^{II}$ – $Cu^{I}$ redox potential, as demonstrated by the highest discharge potential of the 30 m $ZnCl_2$-based biphasic electrolyte. Such a phenomenon was reported elsewhere[27], which is attributed to an increase in the kinetic rate constant for the redox of $Cu^{II}$ species due to the enhanced mass transport at high chloride concentrations. The discharge capacity of the biphasic electrolyte with higher concentrations of $ZnCl_2$ (15 m and 25 m) is 395 and 388 mAh $g^{-1}$ based on the mass of $CuCl_2$, respectively, indicating $CuCl_2$ was fully converted with two-electron transfer (theoretical capacity of 398 mAh $g^{-1}$). EDS analysis on the zinc surface after cycling in the 15 m $ZnCl_2$-based biphasic electrolyte confirms free Cu deposition, indicating suppressed Cu species crossover (Supplementary Fig. 3). The cycling stability and the coulombic efficiency are also strongly correlated to the $ZnCl_2$ concentration (Fig. 1d), among which the 15 m $ZnCl_2$-based biphasic electrolyte manifests the highest coulombic efficiency with improved stability. The compromised discharge capacity and stability at higher $ZnCl_2$ concentrations might be attributed to its high viscosity[28,29], because viscosity inflection occurs at approximately 20 m[30].

Schematic illustrations of the comparison between the single-phase electrolyte and biphasic electrolyte are shown in Fig. 1e, f. To complete the chloride ion circuit in the cell, a routine and inexpensive dye 4-[(4-aminophenyl)-(4-imino-1-cyclohexa-2,5-dienylidene)methyl] aniline hydrochloride (fuchsine, 0.25 m, saturated) was added to the organic phase. Decreasing the fuchsine concentration would increase the overpotential of the battery (Supplementary Fig. 4). Due to the hydrophobicity of the aromatic conjugated structure, fuchsine is more likely to remain in the organic phase compared to other general quaternary ammonium chlorides. The room temperature ionic conductivity of the 15 m $ZnCl_2$-based biphasic electrolyte is 1.03 mS $cm^{-1}$, while that of the aqueous phase and organic phase is 18 mS $cm^{-1}$ and 0.49 mS $cm^{-1}$, respectively (Supplementary Fig. 5, biphasic electrolyte #2). The influence of the thickness ratio of the aqueous/organic phase on the conductivity was also considered (Supplementary Fig. 5b). Since the ionic conductivity of the aqueous phase is greater than that of the organic phase, the ionic conductivity increases with increasing

thickness ratio of the aqueous/organic phase. Supplementary Fig. 6 shows the linear sweep voltammetry (LSV) curve for the organic phase of the biphasic electrolyte. There is a small fraction of water in the organic phase; however, the hydrogen evolution potential is extended to −1.8 V vs. SHE.

The above results evidenced that a proper "ion-selective" interface of the biphasic electrolyte could prevent $Cu^{II}$ shuttling; thus, a high coulombic efficiency and reversibility of the Zn–Cu cell could be achieved. By using the same battery configuration described in Fig. 1f, we successfully expanded the Zn–Cu chemistry to iron chloride, nickel chloride, and vanadium oxides as cathode active materials. These metal ions are highly soluble in aqueous solution and thus involve similar crossover issues; however, they are properly controlled by the biphasic electrolyte, as demonstrated by their stabilized cycling performance (Fig. 1g and Supplementary Fig. 7). These batteries exhibited attractive specific capacities between 1.2 and 1.7 V ($M^0$ states are not formed in this voltage range[31,32]), especially for the Ni–Zn and Fe–Zn chemistries with moderate reversibility and abundance of the active materials. We further expanded the anode to Li metal, which was immersed in the organic phase of the electrolyte to provide a higher discharge potential to the battery (Supplementary Fig. 7g). The discharging voltage of the Li–Cu battery is improved to 3.4 V by using the $Cu^{II}$ – $Cu^{I}$ redox reaction; however, due to the occurrence of a small fraction of $H_2O$ in the organic phase that corrodes the Li metal anode, the battery is mostly irreversible.

## Distribution of $Cu^{2+}$ ions in the biphasic electrolyte and the significance of $ZnCl_2$

Although the $CuCl_2$ active material was blended with super P in the cathode, contact with the aqueous phase of the biphasic electrolyte led to partial dissolution. This dissolution is the major crossover pathway that accounts for the fast capacity decay of the Zn–Cu cell with the neat $ZnCl_2$ electrolytes. However, it is significantly suppressed by the combination of an organic phase that forms the biphasic electrolyte, as discussed above; as a result, the solvation structure of $CuCl_2$ in $ZnCl_2$ aqueous solution is the main influencer for the restriction of copper ions in the aqueous phase. Given that the solubility of $CuCl_2$ in the aqueous phase is affected by the concentration of $ZnCl_2$, here, we chose 1 m $CuCl_2$ in various $ZnCl_2$ solutions to mimic the dynamic electrolyte environment. This selection considered either the saturated concentration of $CuCl_2$ in 20 m $ZnCl_2$ (~1 m at 25 °C) or the ratios between the cathode and the aqueous phase of the biphasic electrolyte. $CuCl_2$ (1 m) in various $ZnCl_2$ solutions was contacted with the [bmmim][$Tf_2N$] ionic liquid, which showed distinct phase separation, as exemplified by the $CuCl_2$–15 m $ZnCl_2$-based biphasic electrolyte (inset of Fig. 2a). The equilibrated copper ions in the organic phase were quantified by ICP–OES tests, showing a decreased copper population with increasing $ZnCl_2$ concentration (Fig. 2a). The combination of 1 m $CuCl_2$ and 15 m $ZnCl_2$ aqueous solution results in minor copper ions in the organic phase. However, 0.5 m $CuCl_2$ without $ZnCl_2$ affords ~115 mg/100 mL in the organic phase, which indicates the significance of $ZnCl_2$ for ion separation in the biphasic electrolyte.

Notably, cations in the ionic liquid phase also have a considerable impact on the distribution of copper ions in the two phases. After equilibration was performed with the 1 m $CuCl_2$–15 m $ZnCl_2$ aqueous solution, the concentrations of $Cu^{2+}$ in the organic phase are in the order of [hmim][$Tf_2N$] > [dmim][$Tf_2N$] > [bmim][$Tf_2N$] > [bmmim][$Tf_2N$] (Fig. 2b). Such a sequence is literally correlated to the hydrophilicity of the cations, in which the C2 methylated [bmmim] shows the best hydrophobicity due to inhibition of the hydrogen bond between water and C2-H[33]. The $Cu^{2+}$ distribution ratio (aqueous concentrations: ILs) is reported in Fig. 2c and Supplementary Fig. 8, showing the high distribution ratio of the [bmmim][$Tf_2N$]-based biphasic electrolyte. The occurrence of the ionic liquid in the aqueous phase was investigated via Raman spectra and $^{19}F$ NMR. The Raman band at

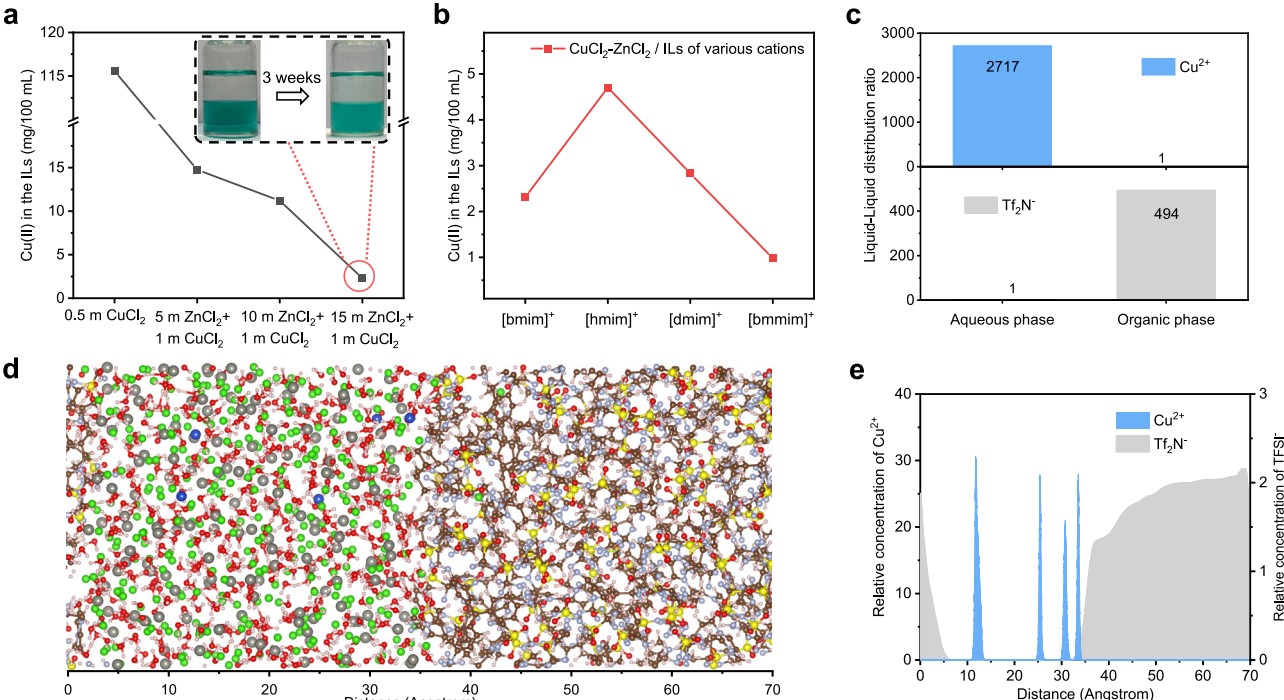

**Fig. 2 | Location of Cu²⁺ ions in the self-stratified aqueous-organic biphasic system. a** The concentration of Cu²⁺ in the [bmmim][Tf₂N] phase after equilibration with the CuCl₂-ZnCl₂ solutions. The inset shows the stratification of the 1 m CuCl₂−15 m ZnCl₂-based biphasic electrolyte. **b** The influence of the cations on the Cu²⁺ population in the organic phase. **c** Density profiles for Cu²⁺ and Tf₂N⁻ obtained by ICP–OES and NMR, respectively. **d** A snapshot extracted from the simulation of the aqueous CuCl₂-ZnCl₂ solution/IL system. The green atom is Cl⁻, the blue atom is Cu²⁺, the red atom is O, the pink atom is H, the powder blue atom is N, the brown atom is C, and the yellow atom is S. **e** Density profiles for Cu²⁺ and Tf₂N⁻ at the interface of the biphasic electrolyte obtained by molecular dynamics simulations.

approximately 740 cm⁻¹ is assigned to the C-F vibration of the ionic liquid; however, it is absent in the aqueous phase of the biphasic electrolyte, indicative of immiscibility (Supplementary Fig. 9). Moreover, only a trace amount of Tf₂N⁻ ($6.73 \times 10^{-3}$ mmol g⁻¹) in the aqueous phase was detected by ¹⁹F NMR, corresponding to a distribution ratio of 494 between the organic phase and the aqueous phase (Fig. 2c and Supplementary Fig. 10). Due to the highly hydrophobic nature of the fuchsine cation, its chloride counter anions are mostly restrained in the organic phase to maintain the charge neutrality. The chloride ion concentration was determined to be 0.23 mol L⁻¹ in the equilibrated organic phase by the silver nitrate titration method, which is sufficient to accomplish the chloride shuttle.

Molecular dynamics (MD) calculations also confirmed the stability of the interface of the biphasic system within a nanoscale, in which the aqueous phase and ionic liquid are stratified along the axis with a length of 7 nm in the simulation box. The calculations illustrate a sharp interface and inerratic distribution of ions between the two phases (Fig. 2d). Although only a rough density profile is provided due to the limited simulation capacity (Fig. 2e), the tendency is consistent with the experimentally observed distribution patterns of Cu²⁺ and Tf₂N⁻ in the two phases. The density profiles for fuchsine and [bmmim]⁺ are also readily available in the organic phase, which again indicates that organic cations are virtually absent in the aqueous phase (Supplementary Fig. 11).

Knowledge of the local coordination environment around Cu²⁺ in aqueous solutions is essential for understanding the distinct distribution of Cu species among the biphasic electrolyte. Classic computations and experiments validated that Cu²⁺ are solvated hydrates ([Cu(H₂O)₆]²⁺) with a distorted octahedral structure at low chloride concentrations, whereas at high concentrations [CuCl]⁺, [CuCl₂] and [CuCl₄]²⁻ coordinated with H₂O ligands are dominant species (copper−water−chloro complexes)[34]. This is due to the stronger dipole moment-derived electronegativity of Cl⁻ than that of H₂O[35], which

induces the displacement of the aqua ligands at the axial position of [Cu(H₂O)₆]²⁺ by Cl⁻ at high chloride concentrations[36]. We experimentally observed this speciation by Raman spectroscopy (Fig. 3a and Supplementary Fig. 12); however, a significant amount of [Cu(H₂O)₆]²⁺ hydrates remained in the 5 m CuCl₂ solution (440 cm⁻¹)[37]. While the solubility of CuCl₂·2H₂O beads in the ionic liquid was significantly lower than that of anhydrous CuCl₂ (Supplementary Fig. 13), it is reasonable to surmise that the copper−water−chloro complexes were the main descriptor inhibiting the solubility of copper ions in the organic phase. Solid copper chloride dihydrate is basically composed of [CuCl₄(H₂O)₂]²⁻ units[38,39], which is similar to the composition of distorted octahedron CuCl₄²⁻ dimers with two aqua ligands in concentrated CuCl₂ aqueous solution[40].

We tested this hypothesis by carrying out additional DFT calculations, in which the conversion of copper-water-chloro complexes and [Cu(H₂O)₆]²⁺ hydrates to Cu(H₂O)₂(Tf₂N)₂ was thermodynamically considered to mimic the crossover of copper ions into the organic phase. The spontaneity of the Cuᴵᴵ cluster shuttle can be quantified by the Gibbs free energy changes[41,42]. Figure 3c illustrates an opposite tendency for the dissociation of [Cu(H₂O)₆]²⁺ and [CuCl₂(H₂O)₄] in the organic phase, suggesting that the former is a spontaneous process and the latter is thermodynamically unfavorable. This hypothesis was also experimentally verified by the CuCl₂-based biphasic electrolyte without ZnCl₂, where both the 0.5 m CuCl₂ and 5 m CuCl₂ aqueous solutions show good stratifications with IL (Supplementary Fig. 14); however, their combination still triggers crossover of copper ions in the Zn−Cu cells due to the presence of copper hydrates (Fig. 3a).

On the other hand, the combination of ZnCl₂ solution as the supporting aqueous phase could significantly suppress the formation of ([Cu(H₂O)₆]²⁺) clusters but was prone to copper-water-chloro complexes. In pure ZnCl₂ solution, the intense Raman band at approximately 283 cm⁻¹ represents [ZnCl₂₊ₓ(H₂O)ᵧ]ˣ⁻ hydrates (Fig. 3b)[30], while the peak at 384 cm⁻¹ is assigned to [Zn(H₂O)₆]²⁺ [43,44]. The [ZnCl₄]²⁻

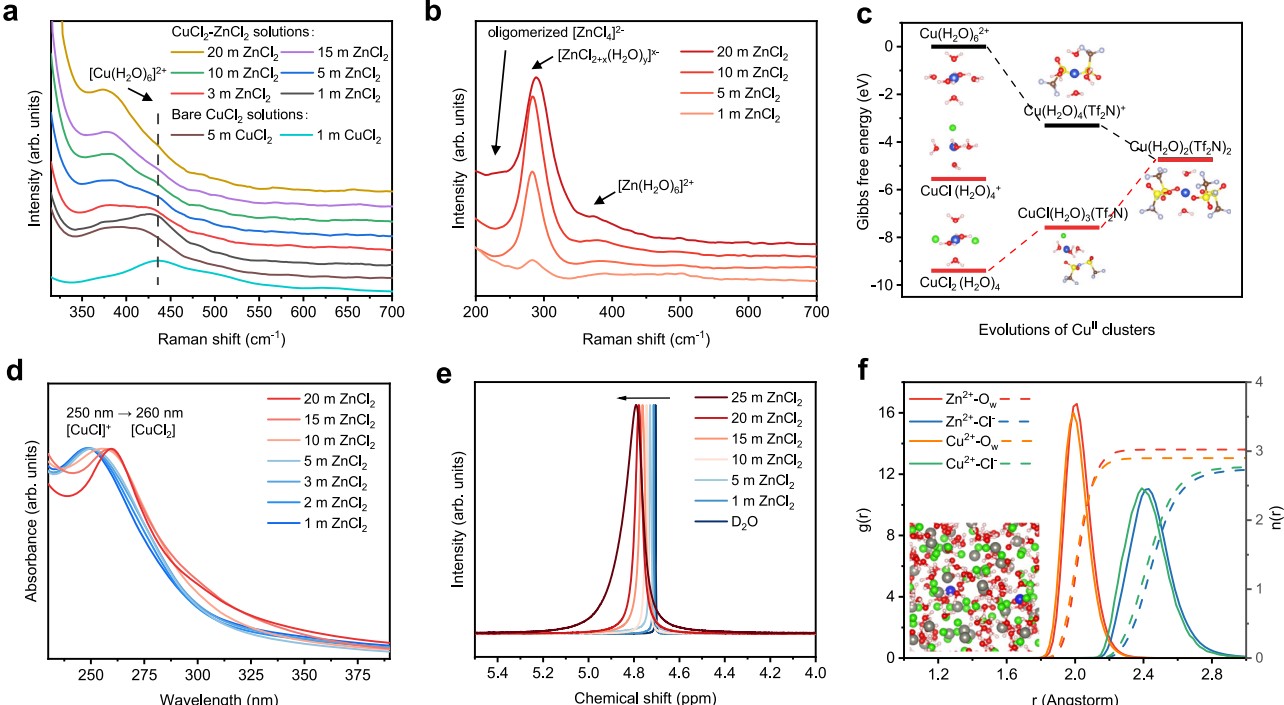

**Fig. 3 | The solution structure of the aqueous phase and its correlation to the Cu distribution. a** Raman spectra of bare CuCl₂ aqueous solution and 1 m CuCl₂ in various ZnCl₂ aqueous solutions. **b** Raman spectra of ZnCl₂ solutions. **c** The Gibbs free energy change for the evolution of Cu$^{II}$ clusters into the organic phase. **d** UV–vis absorbance of copper ions in ZnCl₂ solutions to demonstrate the coordination number of copper-chloro complexes. **e** $^1H$ NMR spectra of ZnCl₂ aqueous solutions. **f** Radical distribution function (RDF) of the CuCl₂–15 m ZnCl₂ aqueous solutions. The solid lines are the radial distribution functions, and the dotted lines are the coordination numbers. The inset is a snapshot of the molecular dynamics simulation box for the aqueous CuCl₂-ZnCl₂ solution.

clusters rapidly dominate the solution with increasing ZnCl₂ concentration (239 cm$^{-1}$), indicating the share of chloride corners of the tetrahedral [ZnCl₄]$^{2-}$ aggregates[45–47]. Although the Raman bands of chloride-associated zinc hydrates and that of copper coincide in the CuCl₂-ZnCl₂ solution, the solvation structure of copper can roughly be described by the presence of complete hydration ([Cu(H₂O)₆]$^{2+}$) clusters. The copper hydration structure at 440 cm$^{-1}$ gradually weakened with increasing ZnCl₂ concentration and vanished in concentrated ZnCl₂ solutions (>10 m) (Fig. 3a and Supplementary Fig. 15). This indicates that H₂O around Cu$^{2+}$ is replaced with Cl$^-$; thus, copper-water-chloro complexes are formed at high ZnCl₂ concentrations. Moreover, the activity of water is suppressed with increasing ZnCl₂ concentration. The Raman band at approximately 3400 cm$^{-1}$ is blueshifted with increasing ZnCl₂, indicating a strengthened O-H bond (Supplementary Fig. 16). The deshielding effect in $^1H$ NMR suggests that the strong interaction of the ion cluster and water breaks the hydrogen bond network (Fig. 3e), which is anticipated to overcome the side reaction of H₂O.

UV–vis absorption spectra were further obtained to analyse the solvation environment of the CuCl₂-ZnCl₂ solutions. Because the detection limit for Cu$^{2+}$ is less than 0.05 m for UV spectroscopy (Supplementary Fig. 17), we controlled the concentration of Cu$^{2+}$ to $3 \times 10^{-3}$ m in the mixed solution. The [CuCl(H₂O)ₓ]$^+$ at 250 nm and [CuCl₂(H₂O)ₓ] at 260 nm are dominant copper-water-chloro complexes in low concentration (5 m) and high concentration (15 m) ZnCl₂ solutions[35], respectively (Fig. 3d). The coordination number of copper–chloro complexes in aqueous solution is dependent on the ratio of copper ion concentration to free Cl$^-$ concentration[48], suggesting the sharing of chloride from the concentrated ZnCl₂ for the formation of dichloride copper structure. This structure was further validated by MD simulations, showing that the copper–water–chloro complexes contain 2.7 chloride and 2.9 H₂O molecules (Fig. 3f). These analyses suggest that instead of being completely hydrated, a

chloride-rich hydration sheath of copper is formed in the CuCl₂-ZnCl₂ solution with high ZnCl₂ concentrations (>10 m). We further provide other zinc salts as control samples in the aqueous phase to highlight the critical role of chloride ions. The aqueous solution of ZnSO₄, Zn(ClO₄)₂, and Zn(NO₃)₂ shows distinct phase separation with the organic phase, forming a biphasic electrolyte solution as the ZnCl₂ salt (Supplementary Fig. 18a). Nonetheless, these anions possess a poor coordinating ability due to their steric hindrance and extensive charge delocalization[49]; thus, all three aqueous solutions are blue, indicative of the presence of a large number of hydrated copper ions. Accordingly, the capacity of the Zn–Cu battery with 4 m ZnSO₄, 5 m Zn(ClO₄)₂, and 5 m Zn(NO₃)₂-based biphasic electrolytes decays rapidly (Supplementary Fig. 18), emphasizing the key role of chloride ions in eliminating the crossover problem in the Zn–Cu battery.

In agreement with this discussion, the successful working of NiCl₂ and FeCl₃ electrodes in the ZnCl₂-based biphasic electrolyte could also be attributed to the suppression of completely hydrated metal clusters in a high concentration ZnCl₂ solution[43,50], as shown in Supplementary Fig. 19. ICP–OES tests confirm that the concentrations of Fe$^{3+}$ and Ni$^{2+}$ in the organic phase are 9.81 and 7.78 mg/100 ml, enabling distribution ratios of 275 and 370 between the aqueous phase and organic phase, respectively. To the best of our knowledge, this is the first work to realize the separation of transition metal ions in two immiscible electrolyte solutions, which is literally governed by the solvation structure of the ions, as discussed above.

**Determining the chemistry of the chloride shuttle-based Zn–Cu battery**

To illustrate the stepwise redox mechanism of copper in the proposed biphasic electrolyte, X-ray diffraction associated with in situ UV–vis spectroscopy was conducted. The UV–vis peaks of copper species in the 15 m ZnCl₂ supporting solution overlap due to the formation of copper (I or II)-water-chloro complexes, i.e., The main peaks of CuCl

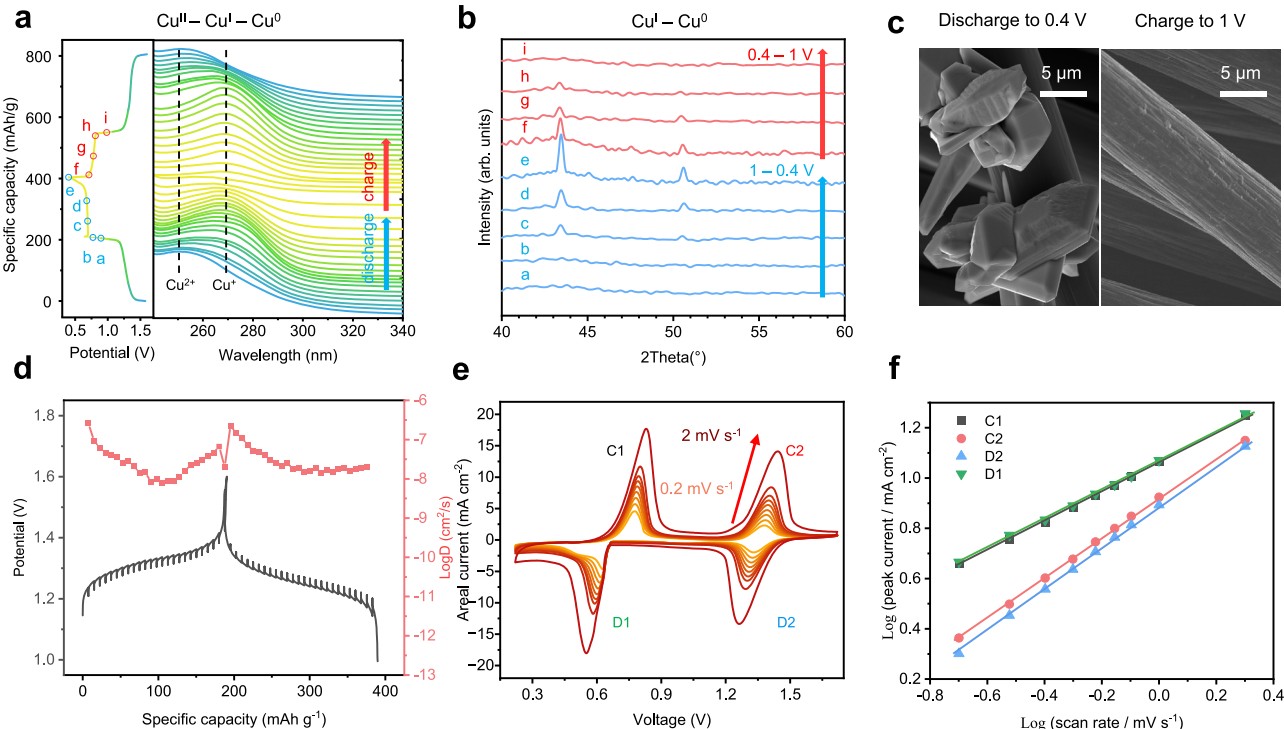

**Fig. 4 | Stepwise redox mechanism and the reaction kinetics of the proposed Zn–Cu battery. a** In situ UV–vis spectra of Cu species in the Zn–Cu battery with the 5 m $ZnCl_2$-based biphasic electrolyte. **b** Ex situ XRD patterns of the cathode at different DODs. **c** SEM images of the Cu deposits on carbon cloth. **d** Diffusion coefficients and voltage profiles of the Zn–Cu battery during GITT measurement. **e** CV curves of Zn–Cu cells with sweep rates of 0.2, 0.3, 0.4, 0.5, 0.6, 0.7, 0.8, 1.0, and 2.0 mV s$^{-1}$. **f** Kinetic analysis of the log *i*-log *v* plots.

and $CuCl_2$ in 15 m $ZnCl_2$ are both located at approximately 260 nm (Supplementary Fig. 20a); however, their positions are adversely correlated with the concentration of the supporting aqueous phase. Specifically, the cuprous-water-chloro complexes are blueshifted with increasing $ZnCl_2$ concentration (Supplementary Fig. 20b), whereas the cupric-water-chloro complexes are redshifted (Fig. 3d). This provides us with the opportunity to qualitatively determine the copper species in a more dilute $ZnCl_2$ solution during the cycling process. Since the redox reaction is not altered by $ZnCl_2$ concentration, albeit in different reversals (Fig. 1d, e), in situ UV–vis spectroscopy was thus conducted in a 5 m $ZnCl_2$-based biphasic electrolyte within a homemade quartz electrochemical cell (Supplementary Fig. 21). Before discharge, the dissolution of $CuCl_2$ into the 5 m $ZnCl_2$ aqueous phase results in a strong UV–vis peak at 250 nm, which is assigned to the cupric-water-chloro complexes (Fig. 4a). Its intensity was weakened and vanished during the first discharge plateau, along with the emergence of cuprous-water-chloro complexes at 260 nm. Further discharge of the cell leads to progressive reduction of $Cu^I$ at the second discharge plateau. The charge process reveals the recovery of $Cu^I$ and its conversion to $Cu^{II}$ for the two-charge plateau.

Posttest analysis of the cathode was further conducted by X-ray diffraction to check the solid product of the cathode. The electrodes at various DODs were rinsed with water prior to the tests. Copper is formed at the second discharge plateau, and its intensity reaches a maximum at the end of discharge; thereafter, the Cu diffraction pattern gradually disappears during the charge process (Fig. 4b). In combination with the UV–vis spectroscopy analysis, the lower voltage plateau is assigned to the conversion between $Cu^I$ and $Cu^0$. We collected the copper particles at the end of discharge by using a carbon cloth current collector with 0.5 m $CuCl_2$ dissolved in the biphasic electrolyte. Figure 4c and Supplementary Fig. 22 show that the Cu polyhedrons are randomly nucleated on the surface of the carbon cloth. The poor contact between the Cu polyhedrons and carbon fibers

indicates that nonpolar carbon is not an ideal substrate for the heterogeneous nucleation of metallic copper.

The chemical composition of the zinc anodes in both discharged and charged states was analyzed by X-ray photoelectron spectroscopy. The high-resolution Cl 2*p* and N 1*s* spectra show the emergence of $ZnCl_2$ (199.3 eV) after discharge, while those of the N signal from the absorbed $Tf_2N$ anions remain unchanged compared to the charged electrode (Supplementary Fig. 23). This confirms that the electrochemical process of the zinc anode in the proposed biphasic electrolyte could be interpreted as zinc plating during charge and conversion to $ZnCl_2$ during discharge. This is because the preferential combination of $Zn^{2+}$ and $Cl^-$ in the organic phase significantly decreases the solubility of $ZnCl_2$ in the organic phase compared to that of $Zn(Tf_2N)_2$ (Supplementary Fig. 24). Since the chloride ions are initially constrained by the organic Fuchsine cation in the organic phase to maintain electroneutrality, the formation and decomposition of solid zinc chloride on the anode during the discharge/charge process involves the shuttling of chloride ions across the interface of the biphasic electrolyte. The transference number of chloride ions ($t_{Cl^-}$) across the interface of the biphasic electrolyte was accessed from the Bruce–Vincent–Evans equation based on chronoamperometry of the symmetric cell using the Ag/AgCl electrode[51]. The Ag/AgCl electrode is a chloride active electrode, which shows stable $Cl^-$ absorption/extraction reactions in the biphasic electrolyte (Supplementary Fig. 25). The $t_{Cl^-}$ was calculated to be 0.38, which is much higher than that of the $t_{Zn^{2+}}$ (0.011) collected by the same method with the zinc symmetric cell. This indicates that chloride ions are the main charge carriers across the interface of the biphasic electrolyte. Moreover, we use an ion exchange membrane to separate the biphasic electrolyte (Supplementary Fig. 26). As expected, the Zn–Cu battery with a cation exchange membrane that breaks the chloride migration failed to cycle, whereas the cell with an anion exchange membrane that allows chloride ion migration demonstrated reversible redox of the $Cu/Cu^{2+}$

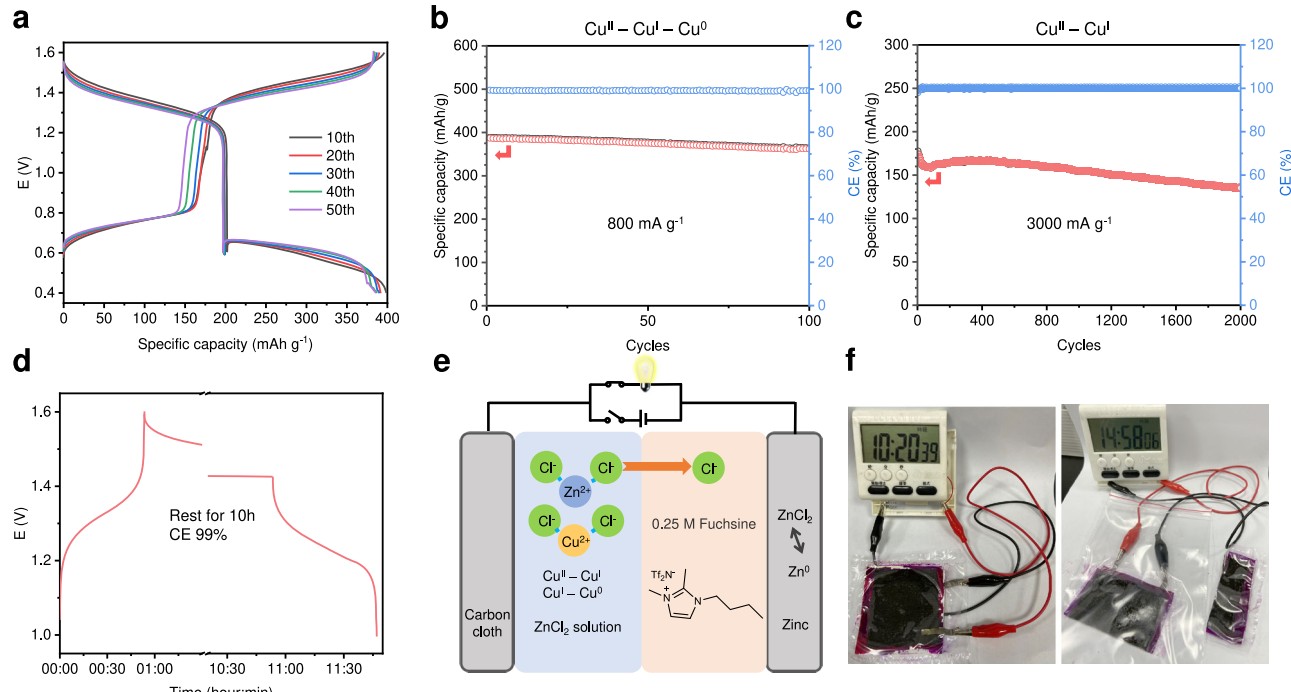

**Fig. 5 | The performance of the proposed Zn–Cu battery. a** Voltage profiles of the Zn–Cu cell between 1.6 and 0.4 V at 400 mA g⁻¹. Cycling performance of the Zn–Cu cell **b** between 1.6 and 0.4 V at 800 mA g⁻¹ and **c** between 1.6 and 0.9 V at 3000 mA g⁻¹. **d** Self-discharge test. The fully charged cell was rested for 10 hours before the discharge process was started. **e** Schematic illustration of the rechargeable Zn–Cu battery for the flexible cell configuration. **f** The flexible Zn–Cu battery powering a digital timer before and after cutting.

couple. Based on the above analysis, the chloride shuttle-involved electrochemical reactions of the proposed Zn–Cu cell could be depicted by the following equations:

$$\text{Cathode}: [CuCl_x]^{2-x} + e^- \rightarrow [CuCl_x]^{1-x} (1.3\,\text{V vs. Zn/Zn}^{2+}, \text{step 1}) \quad (1)$$

$$[CuCl_x]^{1-x} + e^- \rightarrow Cu + xCl^- (0.7\,\text{V vs. Zn/Zn}^{2+}, \text{step 2}) \quad (2)$$

$$\text{Anode}: Zn + 2Cl^- \rightarrow ZnCl_2 + 2e^- \quad (3)$$

The galvanostatic intermittent titration technique (GITT) was conducted to assess the reaction kinetics. The quasiequilibrium potentials are approximately 1.3 V for Cuᴵᴵ – Cuᴵ and approximately 0.7 V for Cuᴵ – Cu⁰ conversions, respectively. The liquid phase conversion (Cuᴵᴵ – Cuᴵ) and deposition-dissolution (Cuᴵ – Cu⁰) in the cathode are discussed separately. The diffusion coefficients of the liquid phase conversion are on the order of $10^{-6}$–$10^{-8}$ cm² s⁻¹ (Fig. 4d), while those of the deposition-dissolution process are on the order of $10^{-7}$–$10^{-9}$ cm² s⁻¹ (Supplementary Fig. 27). The CV curves of the Zn–Cu battery in the biphasic electrolyte with sweep rates of 0.2–2 mV s⁻¹ are shown in Fig. 4e. The kinetic analysis adopted from the plots of log i-log ν indicates that both Cuᴵᴵ – Cuᴵ and Cuᴵ – Cu⁰ conversions are diffusion-controlled redox reactions (Fig. 4f)[52]. The capacitive contribution to the total capacity is 21% at a sweep rate of 0.2 mV s⁻¹ and up to 55% at 2 mV s⁻¹ (Supplementary Fig. 28).

## Performance of the rechargeable Daniell cell with a chloride shuttle

The biphasic electrolyte for cycling performance assessment is composed of 30 μL of 15 m ZnCl₂ aqueous solution and 10 μL of organic phase (see methods for details). Furthermore, Zn–Cu batteries in biphasic electrolytes with different aqueous/organic phase ratios were evaluated (Supplementary Fig. 29), indicating that the specific capacity

and cycling stability are not readily correlated to the aqueous/organic ratios. The cycling stability of the Zn–Cu battery in the proposed biphasic electrolyte was first evaluated between 0.4 and 1.6 V at current densities of 400, 800, and 1600 mA g⁻¹ (1 C = 400 mA g⁻¹) (Fig. 5a, b and Supplementary Fig. 30). The successive reduction of Cuᴵᴵ – Cuᴵ – Cu⁰ provides a comparable discharge capacity of 395 mAh g⁻¹ based on the mass of CuCl₂ (or 835 mAh g⁻¹ based on the Cu mass) at a current density of 400 mA g⁻¹, which is higher than that of state-of-art zinc ion batteries, e.g., MnO₂-Zn, VOₓ-Zn, spinel structured oxides-Zn, and Prussian blue-Zn[53]. This corresponds to a remarkable energy density of 380 Wh kg⁻¹. A specific capacity of 385 mAh g⁻¹ is obtained at a high current density of 800 mA g⁻¹, and the capacity retention is calculated to be 93.5% after 100 cycles with an average coulombic efficiency > 99.5% and an energy efficiency of 83.6%. The Zn–Cu battery was tested for more than 300 cycles at 1600 mA g⁻¹ with a capacity retention of 76%. Supplementary Fig. 31 presents a flat zinc anode surface with no obvious dendrite formation after 100 cycles at 800 mA g⁻¹ between 0.4 and 1.6 V; EDS also confirms that there is no copper deposition on the zinc anode. While the close-to-unit coulombic efficiency is indicative of controlled crossover of the copper species in the cell, the capacity fading of this cell at high current density is mostly attributed to the incompatibility of the nonpolar carbon host for the nucleation of the copper polyhedrons. This is evidenced by the "knee" points at the beginning of the second discharge plateau, in which supersaturated Cuᴵ in the aqueous phase is essential to drive the nucleation of copper metal. This overpotential feature to initiate nucleation was well studied for metal electrodeposition and could be manipulated by altering the surface properties of the substrate; however, it is not the scope of this study.

Alternatively, the cell was very stable when it was tested between 0.9-1.6 V with only the conversion of Cuᴵᴵ – Cuᴵ being activated. It delivers a high discharge capacity of 192 mAh g⁻¹ at a rate of 200 mA g⁻¹, corresponding to complete conversion of Cuᴵᴵ to Cuᴵ. The capacity retention is 85.9% after 150 cycles with a coulombic efficiency of 99.9% and an improved energy efficiency of approximately 94.5%

(Supplementary Fig. 32). At high current densities of 1000 and 3000 mA g$^{-1}$, the long-term stability is also confirmed by the low-capacity decay ratio of 20% for over 800 and 2000 cycles, respectively (Fig. 5c and Supplementary Fig. 32). The stable cycling performance is in sharp contrast to that of the Zn–Cu cell using any of the solutions of the biphasic electrolyte separately, again highlighting the promise of the ion-selective interface of the proposed biphasic electrolyte. Figure 5d shows the self-discharge property of the battery after charging to 1.6 V. The OCV dropped to the equilibrium potential of 1.43 V, and a high CE of 99.9% was obtained after a 10 h interval, indicative of minimal self-discharge of the battery. The proposed Zn–Cu battery also affords robust rate capability, as shown in Fig. 5e and Supplementary Fig. 33. The discharge capacities are 199, 195, 193, 186, and 171 mAh g$^{-1}$ for rates of 200, 600, 1000, 2000, and 3000 mA g$^{-1}$, respectively. The voltage profiles at various rates show that the polarization between charge and discharge was only slightly changed for rates less than 1000 mA g$^{-1}$ (Fig. 5d), indicating the excellent reaction/diffusion kinetics of the Cu$^{II}$ – Cu$^{I}$ redox couple in the aqueous phase. Table S1 provides a comparison of the performance between this study and some representative batteries based on Cu chemistry. The solution solvation structure-related ion selective interface of the biphasic electrolyte could not only greatly alleviate the active material crossover but also provide fast conversion kinetics with improved utilization of the active material because the active material species are partially solubilized in the aqueous phase.

We also fabricated a pouch cell to demonstrate its potential for flexible devices according to the battery configuration shown in Fig. 5e. The cell was sealed in a flexible PE bag with carbon cloth as the current collector for the cathode. The active material CuCl$_2$ (1 m) was dissolved in the biphasic electrolyte instead of blended within the electrode (see Methods). As a result of its flexible components, the Zn–Cu pouch is expected to possess a high flexibility. The hydrophilic carbon cloth and lipophilic PP membrane also endow the battery with gravity independence. The flexible device was connected to a multi-function digital clock (Fig. 5f), and the devise exhibited remarkable durability in bending, inversion, and cutting experiments (Supplementary Fig. 34). The Zn–Cu pouch cell was cut into three equal pieces, which could be repackaged and connected in series to power a 3 V LED lamp (Supplementary Fig. 34).

## Discussion

In summary, we have successfully rebuilt the primary Daniell cell as a secondary battery with an aqueous/organic biphasic electrolyte. An ion-selective interface was established that confines copper ions in the aqueous phase instead of a crossover between the cathode and anode, along with chloride ions serving as the charge carrier between the two phases to maintain electrical neutrality. Such an ion-selective interface was enabled by the combination of immiscible ZnCl$_2$ aqueous solution and a Tf$_2$N-based ionic liquid, in which the CuCl$_2$ cathode and zinc anode are located at the aqueous phase and organic phase, respectively. The Zn–Cu cell delivered a reversible capacity of 396 mAh g$^{-1}$ on accounting for 2e$^-$ stepwise conversion (Cu$^{II}$ – Cu$^0$) with nearly 100% coulombic efficiency, or 199 mAh g$^{-1}$ for the Cu$^{II}$ – Cu$^{I}$ conversion with stable cycling performance. The energy density of the rechargeable Zn–Cu was up to 380 Wh kg$^{-1}$, which is competitive among other aqueous zinc ion batteries. We elucidated that the local coordination environment around Cu$^{II}$ in aqueous solutions is essential for preventing the crossover of Cu ions, which is rationally tunable according to the ZnCl$_2$ concentration with the aim of suppressing the complete hydration of Cu ions. It was demonstrated that the copper-water-chloro complexes are the descriptors that inhibit the presence of Cu in the organic phase, which are dominant in the aqueous solution with >15 m ZnCl$_2$; without these complexes, the copper ions are mostly in their hydration states with spontaneity to be solvated in the organic phase. The strategy of eliminating metal ions in the organic phase is

further expanded to iron chloride, nickel chloride, and vanadium oxides, providing a new promising approach and sustainable power source for large-scale energy storage. The merits of the solution solvation structure-related ion-selective interface of the biphasic electrolyte might also be applicable for the design of advanced chloride ion batteries and flow batteries.

## Methods

### The materials preparation

All reagents were purchased from Shanghai Aladdin Biochemical Technology Co. and used as received without any further purification. Aqueous solutions were prepared by molality (mol-salt in kg-solvent). CuCl$_2$ cathode was prepared by mixing 20 wt% Super P carbon, 70 wt% CuCl$_2$, 10 wt% PTFE, then the mixture was compressed onto Ti mesh (100 mesh, 25.6 mg cm$^{-2}$, 0.2 mm, Hebei Qingyuan Technology Co.). The electrodes of iron chloride, nickel chloride, and vanadium oxides were prepared in the same procedure. The areal loading of the active material on these electrodes was about 4-5 mg cm$^{-2}$. Ag/AgCl electrode was prepared by mixing 20 wt% Super P carbon, 70 wt% AgCl, 10 wt% PVDF, with NMP as the solvent. The slurry was cast onto the Ag foam (28.3 mg cm$^{-2}$, Kunshan Jiayisheng Electronics Co.) and was vacuum dried at 60 °C for 12 h. The areal loading of AgCl is about 20 mg cm$^{-2}$ for each electrode. Carbon cloth (CeTech W0S1011), PP membrane (Celgard 2300), Glass fiber (Whatman GF/A), anion exchange membrane (Fumasep FAB-PK-130), and cation exchange membrane (Nafion N-117) were purchased from sci materials hub. Nafion N-117 was pretreated with 5 wt% H$_2$O$_2$ solution at 80 °C for 1 h to remove the organic impurities, then soaked in the deionized water at 80 °C for 1 h to remove the H$_2$O$_2$. Both anion exchange membrane and cation exchange membrane were pre-soaked in the 15 m ZnCl$_2$ solution before the battery assembly. Carbon cloth was chemically activated by the oxidation method by soaking carbon cloth in a mixture of acids (H$_2$SO$_4$: HNO$_3$ = 1:3) for 12 h.

### Materials characterization

ICP-OES results were conducted on a Varian Agilent 720ES spectrometer. The biphasic electrolytes were equilibrated for a least one week before the ICP tests. UV–vis spectra characterization was carried out on a UV1902PC with a range from 200 to 600 nm. Raman analyses were carried out on a bench Raman dispersive microspectrometer (InVia Reflex, Renishaw) using a laser (wavelength of 532 nm) at frequencies from 100 to 4000 cm$^{-1}$. All the solutions were sealed in the capillary for Raman tests. 1 wt% HCl was added to the aqueous phase in case of coprecipitation of paratacamite at high CuCl$_2$ concentration. XPS spectra were collected on Thermo Scientific K-Alpha system with a monochromatic Al-Ka (1486.6 eV) X-ray source to investigate the chemicals on the Zn anode. $^{19}$F NMR spectra were monitored by a Bruker Ascend HD 400 MHz with deuterium oxide as NMR solvent and trifluoroacetic acid as internal standard. XRD measurements were carried out on a Bruker D8-Advance powder X-ray diffractometer operating at 40 kV and 40 mA, using Cu-Kα radiation (λ = 0.15405 nm). SEM studies were carried out on a TESCAN MIRA3 field-emission SEM instrument. Electrodes were gently washed with deionized water and dried at 60 °C before the SEM tests.

### Computation

All MD simulations were performed within Forcite Package. The COMPASS force-field was used. The Ewald method and the atom-based method were employed for analyzing the Coulomb interactions and the van der Waals (VDW) interactions. In order to obtain a reasonable interaction configuration, a geometry optimization using smart method which is a cascade of the steepest descent, ABNR, and quasi-Newton methods with an energy convergence criterion of 2.0 × 10$^{-5}$ kcal mol$^{-1}$ and force convergence criteria of 1.0 × 10$^{-3}$ kcal mol$^{-1}$ Å$^{-1}$ was used to get a global minimum energy configuration. To further equilibrate the model, the simulations were initially relaxed

under the constant pressure and the constant temperature (NPT ensemble) for 1 ns at a room temperature of 298.15 K and atmospheric pressure. During the simulation, Nose thermostat and Berendsen barostat algorithm were applied in the temperature and pressure control. Later, the equilibrated simulations run at constant NVT ensemble for 10 ns in order to get authentic data. All DFT calculations were performed using the DMol package. The treatment of core electrons was described by the all-electron relativistic method which is the most accurate and also the most computationally expensive of the available type. The generalized gradient approximation of Perdew–Burke–Ernzerhof (GGA–PBE) was used to account for the exchange-correlation functional. The double numerical plus polarization (DNP) basis was used for the best accuracy but highest cost. The global orbital cutoff was set to 4.5 Å. The Grimme correction method was employed in order to include VDW interactions. The energy, force and displacement convergence criterion was set to $1.0 \times 10^{-5}$ Hartree, $2.0 \times 10^{-3}$ Hartree Å$^{-1}$ and $5.0 \times 10^{-3}$ Å for optimization.

## Battery assembly
All electrochemical studies were conducted in Swagelok-type cells with titanium rod current collectors. The diameter of electrodes are 12 mm. The 0.1 mm thickness zinc foil (71.6 mg cm$^{-2}$, 0.1 mm, Hebei Qingyuan Technology Co.) was used as anodes. Glass fiber (Whatman GF/A, 12 mm in diameter) was used as the separator for the cells with the single-phase electrolyte. For the batteries based on biphasic electrolyte, a hydrophilic glass fiber near the cathode was wetted by the ZnCl$_2$ aqueous solution, and a hydrophobic PP membrane (Celgard 2300, 12.6 mm in diameter) near the anode was wetted by the organic solution. The battery was assembled by simply stacking the electrodes and the wetted separators layer by layer. The different hydrophilies of the separators facilitate the combination of two immiscible liquids into a robust biphasic system. The aqueous solution of the biphasic electrolyte for each cell was 30 μL, while that of the organic phase was 10 μL. The Zn–Cu batteries based on the bi-phasic electrolyte with the anion or cation exchange membranes were assembled in the same procedure, except that the anion exchange membrane (Fumasep FAB-PK-130), or cation exchange membrane (Nafion N-117) was placed between 30 μL organic phase and 10 μL aqueous phase. The flexible pouch cell (size: $6 \times 6 \times 0.06$ cm$^3$) was readily fabricated in a transparent PE bag. The biphasic electrolyte for the pouch cell was composed by 800 μL aqueous solution with 1 m CuCl$_2$ and 200 μL organic phase. Hydrophilic carbon cloth with a thickness of 0.36 mm (CeTech W0S1011) was used as the cathode current collector and also as the aqueous phase absorbent, and a hydrophobic PP membrane (Celgard 2300) near the anode was wetted by the organic solution. Hydrophilic glass fiber membrane was not used in the pouch cell. More details please see Supplementary Note 1.

## Electrochemical measurements
Cells were galvanostatically tested on a Neware CT-4008T battery test system (Shenzhen, China) at room temperature (25 °C). Cyclic voltammetry and linear sweep voltammetry measurements were performed on an Interface 1010 electrochemical workstation (Gamry, America) in a three-electrode cell with the Ag/AgCl reference electrode. The electrochemical stability windows (ESW) of [bmmim][Tf$_2$N] and biphasic electrolyte were evaluated using linear sweep voltammograms on nonactive titanium electrodes. The scanning rates was set at 0.5 mV s$^{-1}$ if not specified. EIS tests were conducted with amplitude of 0.01 V and frequency from 0.1 Hz to $1 \times 10^6$ Hz. The ionic conductivity of the electrolyte system was determined by the following equation:

$$\delta = \frac{d}{R_b S} \tag{4}$$

Where d is the distance between electrodes, R$_b$ is the impedance, S is the contact area between electrolytes and titanium disk (diameter = 0.5 inch). In consistent with the structure of the battery, separators were used as absorbent for the electrolyte solution (hydrophilic glass fiber for the ZnCl$_2$ aqueous solution and hydrophobic PP membrane for the organic solution. In GITT test, the cells were performed with the voltage range of 1–1.6 V vs. Zn/Zn$^{2+}$ at 200 mA g$^{-1}$. The duration time for each applied galvanostatic current was 2 min followed by a 20 min relaxation. Ion transference number ($t_{Zn2+}$ and $t_{Cl-}$) was measured with combination measurements of alternating current impedance and direct current polarization using the Zn//Zn and Ag/AgCl//Ag/AgCl cells based biphasic electrolyte, respectively. The polarization currents of cell including initial (I$_0$) and steady-state (I$_{SS}$) were recorded under a direct current polarization voltage of 20 mV (ΔV). The interfacial resistances before (R$_0$) and after (R$_{SS}$) polarization were tested by EIS. Afterwards, ion transference number was calculated from Bruce–Vincent–Evans Equation:

$$t = \frac{I_{SS}(\triangle V - I_0 R_0)}{I_0(\triangle V - I_{SS} R_{SS})} \tag{5}$$

## Data availability
The authors declare that all the relevant data are available within the paper and its Supplementary Information file or from the corresponding author upon reasonable request.

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

## Acknowledgements

X.L. appreciates the support from the National Key Research and Development Program of China (2021YFE0109700, 2019YFA0210600), National Natural Science Foundation of China (51972107), and the Major Program of the Natural Science Foundation of Hunan Province (2021JC0006).

## Author contributions

X.L. conceived and designed the study. C.X. directed the study and contributed materials preparation and electrochemical tests. C.X., J.Y.L., and H.J.W. conducted spectroscopy studies. C.X., T.T.L., and P.J.J. performed zinc electrodeposition and SEM studies. C.J.L. and X.H. conducted MD simulations. C.X. and X.L. contributed to data interpretation

and analysis, and wrote the manuscript. All authors contributed to the scientific discussion

## Competing interests

The authors declare no competing interests.
