## [Peer Review File · Nature Communications]

REVIEWER COMMENTS

Reviewer #1 (Remarks to the Author):

It can be seen that the author has done some careful work to design a simple, long life zinc-copper battery. Although the performance and principle of the battery are discussed in detail, there are still some problems in this paper.

1. The innovation of this paper is slightly insufficient. The open-circuit voltage and discharge platform of this battery are both low.
2. The authors do not elaborate on the role of chloride ions in the cell and suggest a more detailed analysis.

In general, although the paper discusses and analyzes the battery performance and principle at a large length, its innovation is insufficient, and it is suggested that the author should make major revision to the paper.

Reviewer #2 (Remarks to the Author):

I have read the manuscript by Xu et al. I have also read the answer to reviewer's comments. It seems this manuscript was reviewed at another journal and then resubmitted here.

I read the reviewer comments from the other journal and they were very detailed and had similar questions to what I had when I went through this paper independently. The authors have put in considerable effort to answer these questions.

This paper's approach is interesting and the authors do demonstrate with characterization and cycling data that their Zn-Cu system cycles well in this novel electrolyte system. I was astonished that they can get this separation in a real battery or cell configuration. I am afraid they still need to provide more detail for others to repeat these types of interesting cells. What was the weight of Zn anode used? How were the lab cells cycled? Can you put pictures of the full cell design (electrodes, separators, wetted separators, etc) and how the separators were wetted and how long, etc? How is the electrolyte uptake? A side pic of the complete cell before sealing it in the bag would be a great Figure. Where was the materials procured from? This is important. Where was the electrolytes

procured from? Did it need further processing? As this is the main selling point of the manuscript, this needs to be detailed. That is also not given.

The carbon cloth thickness and titanium mesh thickness? Why wasn't titanium mesh used in the prismatic or flexible cell? How was the mix infiltrated into the carbon cloth or was it just surface coated?

Details are important and the authors need to improve on this before acceptance. Other than this I have no issues, the authors have a very interesting concept and have done considerable work to prove this concept and cycling results. Once they improve on the details I will gladly recommend publication.

Reviewer #3 (Remarks to the Author):

The authors present suitable data for demonstrating a Zn-Cu battery that operates in a biphasic electrolyte and did a decent job in replying to previous reviews.

However, the audience for this research is rather niche, where the work does not have the appeal of an all aqueous system and is not competitive with proven organic chemistries.

This paper is better suited for a targeted energy storage journal.

Furthermore the new comparison table is misleading by using current density as the normalizing factor where in this work the loading was ~ 5 mg but in ref 5 the loading was ~ 120 mg. Comparisons should be made by C rate and areal capacity.

C rate should be given along with all currents densities.

Reviewer #1 (Remarks to the Author):

It can be seen that the author has done some careful work to design a simple, long life zinc-copper battery. Although the performance and principle of the battery are discussed in detail, there are still some problems in this paper.

→ We highly appreciate the reviewer's recognition and efforts. These critical comments would definitely help us to further elaborate on the mechanisms of the battery and improve the quality of our manuscript.

1. The innovation of this paper is slightly insufficient. The open-circuit voltage and discharge platform of this battery are both low.

→ We thank the reviewer for this comment. The motivation of this work was to develop a reversible Zn-Cu battery with an efficient approach, which is a competitive energy storage battery due to the abundance of the material. Vast efforts have been dedicated to make the Zn-Cu Daniell battery reversible by using ion-exchange membrane/ceramics (*J. Power Sources*, 2020, 453: 227873; *Chem. Commun.*, 2015, 51(34), 7294–7297), or transferring the redox electrochemistry to hydroxyl (OH⁻) involved precipitation process (*Adv. Funct. Mater.*, 2019, 29(50): 1905979; *J. Power Sources*, 2022, 529: 231168) to mitigate the Cu crossover problem. These approaches either significantly increase the cost of the technology, or result in a large polarization with low utilization of the active materials due to sluggish kinetics of the solid-solid conversion.

Alternatively, we have successfully rebuilt the primary Daniell cell as a secondary battery with an aqueous/organic biphasic electrolyte, which established an ion selective interface that confines copper ion in the aqueous phase instead of being crossover between cathode and anode, along with chloride ion served as the charge carrier between the two phases to keep electrical neutrality. To the best of our knowledge, this is the first work that allows the reversible conversion of metallic species (Cu⁰/Cu²⁺) in a biphasic electrolyte without the support of ion-exchange membrane. Furthermore, we have elucidated that the crossover of the Cu ions across the interface of the biphasic electrolyte is strongly correlated to the coordination structure of Cu ions in the aqueous phase, which is rationally tunable according to the ZnCl₂ concentration. Such a strategy of ion-selective interphase is also expandable to iron chloride and nickel chloride, providing a new promising approach and sustainable power source for largescale energy storage. More importantly, the solution solvation structure related ion selective interface of the biphasic electrolyte in this work could not only greatly alleviate the active material crossover, but also provide a fast conversion kinetics with improved utilization of the active material due to the fact that the active materials species are partially solubilized in the aqueous phase. It might be applicable to the development of future chloride ion batteries and flow batteries.

We agree with the reviewer that the working voltage is an important parameter for energy storage batteries. The proposed Zn-Cu battery shows an open-circuit voltage of 1.55 V, while the discharge platforms are about 1.4 V for Cu^{II} – Cu^I and about 0.7 V for Cu^I – Cu⁰ conversions, respectively. These represent the typical voltages for aqueous batteries, although some recent new battery systems have higher discharge platforms (Zn-MnO₂, 1.85 V; *Adv. Energy Mater.*, 2020, 10(9): 1902085.; four-electron Zn-I₂, 1.65 V; *Nat. Commun.*, 2021, 12(1): 170). For comparison, the state-

of-art zinc ion batteries based on intercalated cathodes have similar discharge voltages, e.g. 1.3 V for MnO₂-Zn (*Nat. Commun.*, 2018, 9(1): 2906) and 0.8 V for VO_x-Zn (*Nat. Energy*, 2016, 1(10): 1). The zinc-iodine battery shows a discharge platform of 1.25 V (*Angew. Chemie*, 2021, 133(7): 3835-3842), while the emerging zinc-sulfur battery only exhibits a discharge platform of 0.5 V (*Adv. Sci.*, 2020, 7(23): 2000761). On the other hand, the reversible capacity of the Cu electrode is as high as 835 mAh g⁻¹ based on the Cu mass, corresponding to a high energy density of 800 Wh kg⁻¹ (or 380 Wh kg⁻¹ based on the CuCl₂ mass), which is competitive among other aqueous zinc ion battery.

2. The authors do not elaborate on the role of chloride ions in the cell and suggest a more detailed analysis.

→ The reviewer arises a valid comment. In addition to the mechanism of the ion selective interface of the biphasic electrolyte we discussed in the manuscript, we have conducted several experiments in the revised manuscript to elaborate on the role of chloride ions. It is summarized that the chloride ions play two essential roles in the proposed Zn-Cu battery: (1) Shuttling between the cathode and anode as a charge carrier; (2) Composing the copper-water-chloro complexes to inhibit the crossover of the copper species.

In order to reveal the migration of ions in the bi-phasic electrolyte, the transference number of zinc ions ($t_{\text{Zn}^{2+}}$) across the interface of the biphasic electrolyte was accessed from the Bruce-Vincent-Evans equation based on the chronoamperometry curve of the Zn/Zn cell (*Adv. Funct. Mater.*, 2022: 2209463) (**Figure R1a, b**). The $t_{\text{Zn}^{2+}}$ is calculated to be 0.011 for the biphasic electrolyte, indicating that Zn²⁺ hardly contributes to ion migration across the interface. The Ag/AgCl electrode was prepared as a chloride active electrode, which shows stable Cl⁻ absorption/extraction reactions in the biphasic electrolyte (**Figure R1c**). It allows us to test the chloride ion transference number (t_{Cl^-}) of the biphasic electrolyte in a symmetric cell (**Figure R1a, d**). The t_{Cl^-} is calculated to be 0.38, which is much higher than that of the $t_{\text{Zn}^{2+}}$. It indicates that chloride ions are the main charge carrier across the interface of the biphasic electrolyte. On the other hand, we use an ion exchange membrane to separate the biphasic electrolyte. As expected, the Zn-Cu batteries in the biphasic electrolyte with anion (Fumasep FAB-PK-130) or cation (Nafion N-117) exchange membrane show distinct performance (**Figure R2**). The battery with a cation exchange membrane that breaks the chloride migration failed to cycle. In sharp contrast, the cell with an anion exchange membrane that allows chloride ion migration demonstrated reversible redox of the Cu/Cu²⁺ couple, which is similar to that of the step-wise redox process of the cell without ion selective membrane. This is valid since the amount of chloride ions in the organic phase (provided by fuchsine, 0.25 m) is insufficient for the Zn/ZnCl₂ conversion. It requires a concentration of the chloride to exceed 3.7 m in 10 μL organic phase to accomplish the 1 mAh cm⁻² areal capacity, which is far beyond the available 0.25 m fuchsine, verifying the migration of chloride ions from the aqueous phase during the charge and discharge process. Moreover, the chemical compositions of the zinc anodes at the discharged and charged states were analyzed by X-ray photoelectron spectroscopy, revealing the conversion between Zn metal and ZnCl₂ (**Figure S22**).

The mitigation of copper crossover by the biphasic electrolyte was comprehensively elucidated in the original manuscript, which is summarized as the correlation between the solution structure of the biphasic electrolyte and the copper ion solvation structure. It demonstrated that the copper-

water-chloro complexes are the descriptor that inhibits the occurrence of Cu in the organic phase, which are dominated in the aqueous solution with $> 15 \text{ M ZnCl}_2$; otherwise, the copper ions are mostly in their hydration states with spontaneity to be solvated in the organic phase. We here further provide other zinc salts as control samples in the aqueous phase to highlight the critical role of the chloride ions (**Figure R3**). The ZnSO_4 , $\text{Zn}(\text{ClO}_4)_2$, and $\text{Zn}(\text{NO}_3)_2$ aqueous solution shows distinct phase separation with the organic phase (**Figure R3a**), forming a biphasic electrolyte solution as the ZnCl_2 salt. Nonetheless, these anions possess a poor coordinating ability due to their steric hindrance and extensive charge delocalization (*Dalt. Trans.* 2011, 40, 10742), thus all of these three aqueous solutions are in blue color indicative of the presence of a large number of hydrated copper ions. In accordance, the capacity of Zn-Cu battery with these biphasic electrolytes decay rapidly (**Figure R3b-d**), emphasizing the key role of chloride ion for the elimination of crossover problem in the Zn-Cu battery.

These are now added in the revised manuscript.

Figure R1 (a) Schematic diagram of the cells used for the chronoamperometry test. The biphasic electrolyte was consisted of $30 \mu\text{L}$ aqueous phase and $10 \mu\text{L}$ organic phase. (b) CA curves of Zn//Zn cell based on the bi-phasic electrolyte with a perturbation potential of 20 mV , and the insets are the electrochemical impedance spectra at initial and steady states, respectively. (c) The galvanostatic Cl^- absorption/extraction reactions in Ag/AgCl//Ag/AgCl cell based on the bi-phasic electrolyte at 0.1 mA cm^{-2} , 0.25 mA cm^{-2} , and 0.5 mA cm^{-2} . Ag/AgCl electrode was prepared by mixing $20 \text{ wt}\%$ Super P carbon, $70 \text{ wt}\%$ AgCl, $10 \text{ wt}\%$ PVDF, with NMP as the solvent. The slurry was cast onto the Ag foam (28.3 mg cm^{-2}) and was vacuum dried at $60 \text{ }^\circ\text{C}$ for 12 h . The areal loading of AgCl is about 20 mg cm^{-2} for each electrode. (d) The CA curves of Ag/AgCl//Ag/AgCl cell under the same condition as that of Zn//Zn cell.

Figure R2 Voltage profiles of the Zn/Cu battery based on the bi-phasic electrolyte with an anion or cation exchange membranes at 80 mAh g⁻¹.

Figure R3 (a) Stratification between [bmmim][Tf₂N] ionic liquid with the aqueous phase of 4 m ZnSO₄ + 1 m CuCl₂, 5 m Zn(ClO₄)₂ + 1 m CuCl₂, and 5 m Zn(NO₃)₂ + 1 m CuCl₂ solution. Fast degradation cells with (b) 4 m ZnSO₄, (c) 4 m Zn(ClO₄)₂, and (d) 5 m Zn(NO₃)₂ as the aqueous phase of the bi-phasic electrolyte.

In general, although the paper discusses and analyzes the battery performance and principle at a large length, its innovation is insufficient, and it is suggested that the author should make major revision to the paper.

→ We would like to thank the reviewer for the helpful suggestions.

Reviewer #2 (Remarks to the Author):

I have read the manuscript by Xu et al. I have also read the answer to reviewer's comments. It seems this manuscript was reviewed at another journal and then resubmitted here.

I read the reviewer comments from the other journal and they were very detailed and had similar questions to what I had when I went through this paper independently. The authors have put in considerable effort to answer these questions.

→ We highly appreciate the reviewer's recognition and efforts. These are truly important and definitely help us to improve the quality of the manuscript. We have made a point-by-point response as shown below. We hope that we have addressed the concerns of the reviewer.

This paper's approach is interesting and the authors do demonstrate with characterization and cycling data that their Zn-Cu system cycles well in this novel electrolyte system. I was astonished that they can get this separation in a real battery or cell configuration. I am afraid they still need to provide more detail for others to repeat these types of interesting cells. What was the weight of Zn anode used? How were the lab cells cycled? Can you put pictures of the full cell design (electrodes, separators, wetted separators, etc) and how the separators were wetted and how long, etc? How is the electrolyte uptake? A side pic of the complete cell before sealing it in the bag would be a great Figure. Where was the materials procured from? This is important. Where was the electrolytes procured from? Did it need further processing? As this is the main selling point of the manuscript, this needs to be detailed. That is also not given.

→ We thank the reviewer for pointing out the omission of the experimental details. The weight of each Zn foil (12 mm diameter, 0.1 mm thickness, Hebei Qingyuan Technology Co.) for the Swagelok cell is about 80.9 mg. All materials for the electrode and the electrolyte solution were procured from Shanghai Aladdin biochemical technology co., which were used without further processing. Carbon cloth (CeTech W0S1011), PP membrane (Celgard 2300), Glass fiber (Whatman GF/A), anion exchange membrane (Fumasep FAB-PK-130), and cation exchange membrane (Nafion N-117) were purchased from sci materials hub. The pictures of the CuCl_2 electrodes, Ag/AgCl electrodes, zinc anode, and separators are shown in **Figure R4**.

Since the biphasic electrolyte is fluidity for both phases, it is more feasible to use the biphasic electrolyte by employing absorbents in the battery to prevent any distribution of the separation. The hydrophobic PP membrane (Celgard 2300, 12.6 mm in diameter) was used as the absorbent for the organic phase (**Figure R5a**), while the hydrophilic glass fiber was used for the aqueous phase (**Figure R5b**). It also shows that the PP membrane could not be wetted by the aqueous phase (**Figure R5c**), whereas both of the glass fiber and carbon cloth could be wetted (**Figure R5d, e**). The contact of these two absorbent layers in the battery could form the biphasic interface automatically.

The battery was assembled by simply stacking the electrodes and the wetted separators layer by layer (**Figure 6a, b**). The complete cell configurations for the Swagelok cell and pouch cell before sealing are shown in **Figure R6c, d, e**. The cells were vertically placed with the anode side (organic phase) on top during the electrochemical testing process.

These are now added in the revised manuscript.

Figure R4 Pictures of the (a) Ti mesh supported CuCl_2 cathode, (b) glass fiber separator, (c) PP separator, (d) Ag/AgCl electrode, (e) zinc anode, and (f) carbon cloth.

Figure R5 Pictures of the (a) PP separator, PP separator absorbed with the organic phase, (b) glass fiber, and glass fiber wetted by the aqueous phase. Note that the organic phase turned red color after 0.25 m fuchsine was added. Pictures of the (c) PP separators, (d) carbon cloth, and (e) glass fiber with the contact of 10 μL aqueous electrolyte.

Figure R6 (a) and (b) Schematic diagrams for the proposed Zn-Cu battery and flexible pouch cell, respectively. (c), (d) a complete cell before sealing in Swagelok cell case. (e) the flexible pouch cell before sealing.

The carbon cloth thickness and titanium mesh thickness? Why wasn't titanium mesh used in the prismatic or flexible cell? How was the mix infiltrated into the carbon cloth or was it just surface coated?

→ The thickness of the carbon cloth and the titanium mesh is 0.36 mm and 0.2 mm, respectively. The electrode for the Swagelok cell was prepared by mixing 20 wt% Super P carbon, 70 wt% CuCl_2 , 10 wt% PTFE, then the mixture was compressed onto Ti mesh. For the flexible cell, the electroactive material CuCl_2 was dissolved in the ZnCl_2 aqueous solution. A high surface area current collector is preferred in this case to ensure a high active material utilization ratio, thus carbon cloth was used instead of the Ti mesh. Besides, the edge of the Ti mesh is sharp and can impale the package during the bend. **Figure R5e** shows that the aqueous phase could penetrate into the carbon cloth. So, the glass fiber that used as aqueous phase absorbent in the Swagelok cell was not needed for the flexible cell (**Figure R6e**).

Details are important and the authors need to improve on this before acceptance. Other than this I have no issues, the authors have a very interesting concept and have done considerable work to prove this concept and cycling results. Once they improve on the details I will gladly recommend publication.

→ We would like to thank the reviewer for his/her positive comments and valuable suggestions. We agree with the reviewer's comment that details are important for the work, and we have added those details in the revised manuscript.

Reviewer #3 (Remarks to the Author):

The authors present suitable data for demonstrating a Zn-Cu battery that operates in a biphasic electrolyte and did a decent job in replying to previous reviews.

→ Thanks so much for the reviewer's insightful comments. We have made point-by-point response as shown below. We hope that we have addressed the concerns of the reviewer.

However, the audience for this research is rather niche, where the work does not have the appeal of an all aqueous system and is not competitive with proven organic chemistries. This paper is better suited for a targeted energy storage journal.

→ We thank the reviewer for his/her comments. However, this work covers several hot topics instead of only on Zn-Cu battery itself. While state-of-the-art Li-ion batteries are approaching their theoretical limitations, the pursuit of new battery chemistry that have the advantages of resources abundance, safety, and long term stability is the eternal target for energy storage batteries. We have successfully rebuilt the primary Daniell cell as a secondary battery with an aqueous/organic biphasic electrolyte, which established an ion selective interface that confines copper ion in the aqueous phase instead of being crossover between cathode and anode, along with chloride ion served as the charge carrier between the two phases to keep electrical neutrality. We hope the reviewer could agree with us that it is a comprehensive study that includes reversible chloride ion battery chemistry (CIB), copper cathode with detailed battery chemistry, and electrolyte solvation chemistry.

The electrochemical performance of the proposed Zn-Cu battery is competitive to these batteries with organic chemistry and the state-of-art aqueous batteries. This work shows successive reduction of $\text{Cu}^{\text{II}} - \text{Cu}^{\text{I}} - \text{Cu}^0$, providing a comparable discharge capacity of 395 mAh g^{-1} based on the mass of CuCl_2 (or 835 mAh g^{-1} based on the Cu mass). It is superior to that of the batteries with organic active materials, which normally have a specific capacity of $< 300 \text{ mAh g}^{-1}$ (*Chem*, 2018, 4(12): 2786-2813.; *Angew. Chemie*, 2020, 59(48): 21293-21303). When comparing to the all aqueous systems, it is also higher than that of the state-of-art zinc batteries e.g. MnO_2 -Zn, VO_x -Zn, spinel structured oxides-Zn, and Prussian blue-Zn (*Chem. Soc. Rev.*, 2020, 49(13): 4203-4219), except for that of the MnO_2 electrode with dissolution-precipitation chemistry. However, the conversion type MnO_2 based battery need to decouple the pH of the aqueous electrolyte with the help of ion selective membranes (*Nat. Energy*, 2020, 5(6): 440-449). On the other hand, the solution solvation structure related ion selective interface of the biphasic electrolyte in this work could not only greatly alleviate the active material crossover, but also provide a fast conversion kinetics with improved utilization of the active material due to the fact that the active materials species are partially solubilized in the aqueous phase.

We would like to further highlight the merits of this work in the following three aspects.

- (1) The chloride conductive interface of the biphasic electrolyte might be applicable for chloride ion batteries. Despite progresses have achieved for CIBs by electrode material development, as demonstrated by metal oxychlorides (*Angew. Chemie*, 2013, 52(51): 13621; *ACS Energy Lett.*, 2017, 2(10): 2341), layered double hydroxides (*Adv. Funct. Mater.*, 2019, 29(36): 1900983.; *Chem. Eng. J.*, 2020, 389: 124376), and chloride ion-doped polymers (*Electrochim. Acta*, 2018,

270: 30-; *ACS Appl. Mater. Interfaces*, 2017, 9(3): 2535), the poor structural stability and limited capacity of these cathode materials still hinder the development of CIB. The conversion type CIB electrodes (metal chlorides) – although with high theoretical capacity – are challenged by the dissolution of the active materials in both aqueous and nonaqueous electrolytes (*Eur. J. Inorg. Chem.*, 2017, 2017(21): 2784-2799). This work reports a rechargeable zinc-copper battery using chloride shuttle in a biphasic electrolyte. Such proposed battery chemistry is expandable to iron chloride, nickel chloride, indicating that the Zn-Cu formulation is not the only choice. This work might attract readers interested in CIBs who aim to develop CIBs with higher energy density and better cycling stability.

- (2) The highly reversible copper chemistry provides a high energy density Zn-Cu battery. The strong solubility of copper ions in both aqueous and organic electrolytes cause severe crossover issue, thus it is regarded as primary cell for a long time. Efforts have been dedicated to make the Zn-Cu Daniell battery reversible, with the use of ion-exchange membrane/ceramics to prevent Cu crossover in the neutral electrolyte (*J. Power Sources*, 2020, 453: 227873.; *Chem. Commun.*, 2015, 51(34): 7294-7297.), or transfer the redox electrochemistry to hydroxyl (OH⁻) or carbonate (CO₃²⁻) involved precipitation process so as to minimize the copper ion dissolution in electrolyte (*Adv. Funct. Mater.*, 2019, 29(50): 1905979.; *Angew. Chemie*, 2022, 61(31): e202203837.). Nevertheless, the ion-exchange membrane approach significantly increases the cost of the technology, while the precipitation strategy results in a large polarization. In this work, the aqueous/organic biphasic electrolyte of the proposed Zn-Cu battery creates an ion selective interface that confines copper ion in the aqueous phase instead of being crossover between cathode and anode. In addition, the copper species are partially solubilized in the aqueous phase of the proposed biphasic electrolyte, which promises fast conversion kinetics with improved mass transfer. The proposed Zn-Cu battery reveals nearly 100 % utilization of the active materials at high current density.
- (3) The solution solvation structure related ion selective interface of the biphasic electrolyte might be applicable for flow batteries. Flow-battery is considered as a reliable energy storage system that stores dissolved electroactive materials in two external tanks and separates catholyte and anolyte with the costly ionic-selective membrane. However, crossover in the membrane is still a challenge for the reversibility and cycling stability (*J. Electrochem. Soc.*, 2019, 166(12): A2536.). In our work, the crossover of copper species is eliminated via the solvation structure engineering, delivering superior capacity than organic chemistries. We demonstrated that the copper-water-chloro complexes are the descriptor that inhibits the occurrence of Cu in the organic phase, which are dominated in the aqueous solution with > 15 M ZnCl₂; otherwise, the copper ions are mostly in their hydration states with spontaneity to be solvated in the organic phase. Such a strategy might be applicable for future flow batteries with the elimination of the costly ion selective membranes.

Furthermore the new comparison table is misleading by using current density as the normalizing factor where in this work the loading was ~ 5 mg but in ref 5 the loading was ~120 mg. Comparisons should be made by C rate and areal capacity. C rate should be given along with all currents densities.

→ We thank the reviewer for this valuable comment. We agree that the C rate and areal capacity are important for a comprehensive comparison. Further development of the Zn-Cu with higher areal

capacity is in progress in our laboratory. **Table S1** is now revised as requested.

Table R1 Comparison of various batteries with Cu chemistry. Ion-exchange membrane is denoted as IEM. The areal capacities are given in square brackets.

Cathode reaction and output voltage	Capacity (mAh g ⁻¹) based on m _{Cu}	Active materials utilization	Cycle life	Capacity retention	Over-potential (V)	Ref
$\text{Cu}^{2+} + 2\text{e}^- \rightarrow \text{Cu}^0$; 0.9 V;	160 [7.25 mAh cm ⁻²]	19% at 2 mA cm ⁻² (2.5C)	Primary battery	/	/	1
$\text{Cu}^{2+} + 2\text{e}^- \rightarrow \text{Cu}^0$; 0.96 V; IEM supported	330 []	39% at 1 mA cm ⁻²	/	/	0.3	2
$\text{Cu}^{2+} + 2\text{e}^- \rightarrow \text{Cu}^0$; 0.8 V; IEM supported	763 [0.5 mAh cm ⁻²]	90% at 0.5 mA cm ⁻² (0.5C)	100	95%	0.35	3
$\text{Cu}(\text{OH})_2/\text{CuO} + 2\text{e}^- \rightarrow \text{Cu}^0$; 0.76 V	718 [1.5 mAh cm ⁻²]	85% at 0.12C	200	55.8%	0.45	4
$\text{CuO} + 2\text{e}^- \rightarrow \text{Cu}^0$; 0.75 V	843 (674 based on m _{CuO}) [81 mAh cm ⁻²]	>99% at 0.2C	150	56.0%	0.3	5
$\text{CuO} + 2\text{e}^- \rightarrow \text{Cu}^0$; 0.8 V	550 []	66% at 0.2C	200	54.5%	0.3	6
$\text{Cu}_2\text{CO}_3(\text{OH})_2 + 4\text{e}^- \rightarrow \text{Cu}^0$; 0.4 V	665 [13.3 mAh cm ⁻²]	81% at 0.6C	50	48.9%	0.3	7
$\text{Cu}^{\text{II}} + \text{e}^- \rightarrow \text{Cu}^{\text{I}}$; 1.3 V $\text{Cu}^{\text{I}} + \text{e}^- \rightarrow \text{Cu}^0$; 0.7 V	814 (385 based on m _{CuCl2}) [2 mAh cm ⁻²]	96% at 2C	100	93.5%	0.15	This work
	791 (374 based on m _{CuCl2}) [1.9 mAh cm ⁻²]	94% at 4C	300	76.0%	0.2	
$\text{Cu}^{\text{II}} + \text{e}^- \rightarrow \text{Cu}^{\text{I}}$; 1.3 V	397 (188 based on m _{CuCl2}) [1 mAh cm ⁻²]	94% at 2.5C	800	80.9%	0.15	This work
	370 (175 based on m _{CuCl2}) [0.9 mAh cm ⁻²]	88% at 7.5C	2000	80.2%	0.2	

References

- Duan, J. *et al.* Tough hydrogel diodes with tunable interfacial adhesion for safe and durable wearable batteries. *Nano Energy* **48**, 569–574 (2018).
- Zhang, H. *et al.* Using Li⁺ as the electrochemical messenger to fabricate an aqueous rechargeable Zn–Cu battery. *Chem. Commun.* **51**, 7294–7297 (2015).
- Jameson, A., Khazaeli, A. & Barz, D. P. J. A rechargeable zinc copper battery using a selective cation exchange membrane. *J. Power Sources* **453**, 227873 (2020).
- Zhu, Q. *et al.* Realizing a rechargeable high-performance Cu-Zn battery by adjusting the solubility of Cu²⁺. *Adv. Funct. Mater.* **29**, 1–8 (2019).
- Schorr, N. B. *et al.* Rechargeable alkaline zinc/copper oxide batteries. *ACS Appl. Energy Mater.* **4**, 7073–7082 (2021).

6. Arnot, D. J., Schorr, N. B., Kolesnichenko, I. V. & Lambert, T. N. Rechargeable alkaline Zn–Cu batteries enabled by carbon coated Cu/Bi particles. *J. Power Sources* 529, 231168 (2022).
7. Gallagher, T. C. et al. From Copper to Basic Copper Carbonate: A Reversible Conversion Cathode in Aqueous Anion Batteries. *Angew. Chemie Int. Ed.* (2022) doi:10.1002/anie.202203837.

REVIEWERS' COMMENTS

Reviewer #4 (Remarks to the Author):

The authors carefully designed a biphasic electrolyte to avoid Cu cross over in the traditional Daniell Cell and have realized rechargeability in the Cu-Zn cell. This work is quite interesting and efforts have been made in unveiling the role of electrolyte compositions (IL cations, aqueous phase concentration, aqueous/organic phase ratios, etc.), coordination structure of Cu-water-chloro complexes and understanding the redox chemistry of the whole cell, flexible pouch cells are also demonstrated. The authors have added on necessary experimental results and explanations to address the concerns raised by the other reviewers. I suggest the article to be accepted after addressing the following concerns.

1. The redox reactions on the cathode are $\text{CuI}-\text{CuI}-\text{CuO}$, with a theoretical capacity of 398 mAh/g. Given the similar molar mass between NiCl_2 and CuCl_2 , the capacity of NiCl_2 and CuCl_2 should be similar. But in Figure 1g, NiCl_2 and FeCl_3 realized obviously lower specific capacity? Is it because Ni^{2+} and Fe^{3+} are not fully reduced to the MO states in the voltage range? The authors might need to provide some explanations on this.

2. Fuchsine was added into IL to assist ZnCl_2/Zn redox, will the concentration of Fuchsine have any effect on the electrochemical performance of this biphasic Zn-Cu battery?

3. In line 92, the authors mentioned manganese chloride but didn't discuss further in the main text.

4. There are some small errors in the article. In line 196, BMMIM should have the best "hydrophobicity" if I'm not wrong. In line 392, it should be "corresponds". In line 393, the unit of current density is incorrect. In line 394, the energy efficiency of 83.6% was reported. How was the energy efficiency calculated?

Reviewer #5 (Remarks to the Author):

Aqueous Zn batteries with intrinsic safe and low-cost merits are drawing great attention recently. Xu and co-workers demonstrated their findings on a reversible Zn-Cu battery system enabled by a ZnCl₂ based biphasic electrolyte. It is good that the authors are looking into such an important topic and the idea of ion separation by optimizing biphasic electrolyte interface is impressive and of interest. Particularly, the dual role of Cl⁻ as the charge carrier and Cu²⁺ crossover preventer has been elucidated by both spectroscopic and computational studies. The conclusions are sound and well-presented. In principle, it can be published in Nature Communications. Please consider the minor points as noted below:

1. I believe the main charge carriers across the biphasic electrolyte interface are Cl⁻. However, the transfer number of Cl⁻ was determined to be only 0.38, much lower than 1, which makes me wonder if there are other charge carriers than Cl⁻. The authors should be careful and double-check these results. Besides, I suggest to include Figure S33 in the main text for better demonstration of the reversible Zn-Cu chemistry.
2. How was the ionic conductivity of the electrolytes measured and evaluated? The details are suggested to be provided in the experimental section.
3. The figures in the main text are of low resolution, please improve them for better readability.

Reviewer #4 (Remarks to the Author):

The authors carefully designed a biphasic electrolyte to avoid Cu cross over in the traditional Daniell Cell and have realized rechargeability in the Cu-Zn cell. This work is quite interesting and efforts have been made in unveiling the role of electrolyte compositions (IL cations, aqueous phase concentration, aqueous/organic phase ratios, etc.), coordination structure of Cu-water-chloro complexes and understanding the redox chemistry of the whole cell, flexible pouch cells are also demonstrated. The authors have added on necessary experimental results and explanations to address the concerns raised by the other reviewers. I suggest the article to be accepted after addressing the following concerns.

→ We would like to thank the reviewer for his/her positive/constructive comments, which have helped us to further improve the quality of our manuscript.

1. The redox reactions on the cathode are $\text{Cu}^{\text{II}}-\text{Cu}^{\text{I}}-\text{Cu}^0$, with a theoretical capacity of 398 mAh/g. Given the similar molar mass between NiCl_2 and CuCl_2 , the capacity of NiCl_2 and CuCl_2 should be similar. But in Figure 1g, NiCl_2 and FeCl_3 realized obviously lower specific capacity? Is it because Ni^{2+} and Fe^{3+} are not fully reduced to the M^0 states in the voltage range? The authors might need to provide some explanations on this.

→ Thank you for your suggestions. As the reviewer suggested, Fe^{III} and Ni^{II} are not fully reduced to the M^0 states in the voltage range. The FeCl_3 and NiCl_2 shows multistep reduction in the chloride ion coordination environment. $\text{Fe}^{\text{III}}/\text{Fe}^{\text{II}}$ couple shows voltage ranged between 0.6 to 0.8 V vs. SHE, and the deposition of Fe^0 shows voltage lower than - 0.5 V vs. SHE (*Electrochimica Acta*, 2015, 154: 462-467; *Journal of the Electrochemical Society*, 1982, 129(11): 2474). NiCl_2 shows a one-electron redox at potential ranged between 0.5 to 0.9 V vs. SHE, and Ni metal deposition voltage lower than -0.4 V vs. SHE (*Transactions of the IMF*, 2008, 86(4): 234-240; *Surface Engineering*, 2017, 33(2): 131-135.). The theoretical capacity of the one-electron conversion of FeCl_3 is calculated to 165 mAh g^{-1} , while that of NiCl_2 is 206 mAh g^{-1} . The reversible discharge capacities of NiCl_2 and FeCl_3 shown in Figure 1c are close to their theoretical capacities, indicating a high utilization of active materials. We have added an explanation for it in the revised manuscript.

2. Fuchsine was added into IL to assist ZnCl_2/Zn redox, will the concentration of Fuchsine have any effect on the electrochemical performance of this biphasic Zn-Cu battery?

→ We thank the reviewer for this valuable comment. Fuchsine was added in the organic phase to accomplish the chloride ion circuit of the cell due to its solubility and hydrophobicity. We further tested the Zn-Cu battery based on biphasic electrolytes with various fuchsine concentrations (**Figure R1**). It shows decreasing the concentration of fuchsine would increase the overpotential of the Zn-Cu battery. These are now added in the revised manuscript.

Figure R1 The overpotential of the Zn-Cu cells based on biphasic electrolyte is increased along with the decrease of fuchsine concentration. The current density was 200 mA g^{-1} (0.5C).

3. In line 92, the authors mentioned manganese chloride but didn't discuss further in the main text.

→ We apologize for the typo, and we have deleted it in the revised manuscript. In fact, the parasitic reaction or self-discharge caused by the shuttle of manganese ion is not as significant as other transition metal ions, as exemplified by the manganese-zinc battery (*Nature communications*, 2017, 8(1): 405.).

4. There are some small errors in the article. In line 196, BMMIM should have the best “hydrophobicity” if I'm not wrong. In line 392, it should be “corresponds”. In line 393, the unit of current density is incorrect. In line 394, the energy efficiency of 83.6% was reported. How was the energy efficiency calculated?

→ Thanks for pointing out the mistakes, these are now corrected in the revised manuscript.

The energy efficiency was determined according to the equation:

$$\eta = \frac{\int_0^{t_0} V_d(t) I_d(t) dt}{\int_0^{t_1} V_c(t) I_c(t) dt} \times 100\% \quad (1)$$

V_d and V_c are the charge/discharge voltage; I_d and I_c are the current; t is the time of charge or discharge. Energy efficiency is equal to the ratio of the integral area of discharging and charging curve. Actually, the energy efficiency could be automatically recorded by the Neware CT-4008T battery test system.

Reviewer #5 (Remarks to the Author):

Aqueous Zn batteries with intrinsic safe and low-cost merits are drawing great attention recently. Xu and co-workers demonstrated their findings on a reversible Zn-Cu battery system enabled by a ZnCl₂ based biphasic electrolyte. It is good that the authors are looking into such an important topic and the idea of ion separation by optimizing biphasic electrolyte interface is impressive and of interest. Particularly, the dual role of Cl⁻ as the charge carrier and Cu²⁺ crossover preventer has been elucidated by both spectroscopic and computational studies. The conclusions are sound and well-presented. In principle, it can be published in Nature Communications. Please consider the minor points as noted below:

→ We would like to thank the reviewer for his/her positive/constructive comments, which have helped us to further improve the quality of our manuscript.

1. I believe the main charge carriers across the biphasic electrolyte interface are Cl⁻. However, the transfer number of Cl⁻ was determined to be only 0.38, much lower than 1, which makes me wonder if there are other charge carriers than Cl⁻. The authors should be careful and double-check these results. Besides, I suggest to include Figure S33 in the main text for better demonstration of the reversible Zn-Cu chemistry.

→ We thank the reviewer for raising this concern. We have repeated the experiment and accessed the same t_{Cl^-} of 0.38 (**Figure R2**), which indicates the validation of the method. Any cations in the electrolyte could migrate across the interface of the biphasic electrolyte under the electric field, including the organic [bmmim]⁺ cation or fuchsine cation and the solvated zinc ions. These cation migration leads to a non-unity transfer number of the chloride ion.

The Li⁺ transfer number of the traditional Li-ion electrolytes is ranged between 0.2 and 0.3 (*Journal of the Electrochemical Society*, 2002, 149(6): A667). It was also reported that an elevated t_{Li^+} of 0.4 was achieved with the use of a Nafion ionic exchange membrane to facilitate cation migration (*Nano Energy*, 2023: 108174). This is very similar in the case of the biphasic system that the ion-selective interface endows a high transfer number of Cl⁻, which is more than an order of magnitude higher than that of Zn²⁺ (0.38 vs. 0.01).

Thanks for reviewer's constructive advice and the Figure S33 has been added to Figure 5 in the revised manuscript.

Figure R2 Reproduction of the CA test for Ag/AgCl//Ag/AgCl cell based on the bi-phasic electrolyte with a perturbation potential of 20 mV, and the insets are the electrochemical impedance spectra at initial and steady states, respectively.

2. How was the ionic conductivity of the electrolytes measured and evaluated? The details are suggested to be provided in the experimental section.

→ We thank the reviewer for the suggestions. The details related to ionic conductivity measurements was provided in the Supplementary Information, these are now included in the experimental section.

3. The figures in the main text are of low resolution, please improve them for better readability.

→ Thank you for pointing it out. The high-resolution figures are now attached to the manuscript submission system.